# Two enhancer binding proteins activate $\sigma^{54}$-dependent transcription of a quorum regulatory RNA in a bacterial symbiont

Ericka D Surrett[1], Kirsten R Guckes[1], Shyan Cousins[1], Terry B Ruskoski[1], Andrew G Cecere[1], Denise A Ludvik[2], C Denise Okafor[1,3], Mark J Mandel[2], Tim I Miyashiro[1,4]*

[1]Department of Biochemistry and Molecular Biology, Pennsylvania State University, University Park, United States; [2]Department of Medical Microbiology and Immunology, University of Wisconsin-Madison, Madison, United States; [3]Department of Chemistry, Pennsylvania State University, University Park, United States; [4]The Microbiome Center, Huck Institutes of the Life Sciences, Pennsylvania State University, University Park, United States

*For correspondence:
tim14@psu.edu

Competing interest: The authors declare that no competing interests exist.

**Abstract** To colonize a host, bacteria depend on an ensemble of signaling systems to convert information about the various environments encountered within the host into specific cellular activities. How these signaling systems coordinate transitions between cellular states in vivo remains poorly understood. To address this knowledge gap, we investigated how the bacterial symbiont *Vibrio fischeri* initially colonizes the light organ of the Hawaiian bobtail squid *Euprymna scolopes*. Previous work has shown that the small RNA Qrr1, which is a regulatory component of the quorum-sensing system in *V. fischeri*, promotes host colonization. Here, we report that transcriptional activation of Qrr1 is inhibited by the sensor kinase BinK, which suppresses cellular aggregation by *V. fischeri* prior to light organ entry. We show that Qrr1 expression depends on the alternative sigma factor $\sigma^{54}$ and the transcription factors LuxO and SypG, which function similar to an OR logic gate, thereby ensuring Qrr1 is expressed during colonization. Finally, we provide evidence that this regulatory mechanism is widespread throughout the *Vibrionaceae* family. Together, our work reveals how coordination between the signaling pathways underlying aggregation and quorum-sensing promotes host colonization, which provides insight into how integration among signaling systems facilitates complex processes in bacteria.

## Editor's evaluation

The authors present a rigorous and valuable study in which they identify the role of the conserved bacterial enhancer binding protein (bEBP) SypG in regulation of the Qrr1 small RNA, a key regulator of Vibrio fischeri bioluminescence production and squid colonization. The research design and methods were convincing and thorough, leading to compelling conclusions that are broadly relevant to both the quorum sensing and Vibrio-squid symbiosis fields.

## Introduction

The overall fitness of an animal often depends on the activities of bacteria that are localized to certain anatomical sites of the host. In many cases, these bacteria are horizontally transmitted among hosts, which means that they are first shed into a reservoir prior to colonizing a new host. The environmental conditions associated with the reservoir are typically different than those encountered on or within

**Figure 1.** Signal transduction network underlying quorum sensing in *Vibrio fischeri*. Quorum sensing in *V. fischeri* depends on autoinducers 3-oxo-C6 HSL (3OC6 HSL), AI-2, and C8 HSL, which are synthesized by LuxI, LuxS, and AinS, respectively. Interaction of C8 HSL or AI-2 with their cognate sensors (AinR and LuxPQ, respectively), results in lower levels of phosphorylated LuxO. Phosphorylated LuxO promotes σ54-dependent transcription of *qrr1*, which encodes the sRNA Qrr1. Qrr1 post-transcriptionally represses LitR, which is a positive regulator of *luxR*. Consequently, quorum sensing inhibits Qrr1 expression, thereby promoting bioluminescence production. Figure generated with BioRender.com.

a host. Therefore, to properly acclimate to an environment, bacteria depend on signal transduction systems that coordinate cellular physiology in response to a vast array of environmental signals and cues. How these signaling pathways facilitate the cellular activities that are pertinent to the complex environments encountered during host colonization remains unclear for most bacteria. Focusing on the connections between different signaling pathways has the potential to fill this knowledge gap and provide insight into how bacteria transition from one environment to another.

The bioluminescent bacterium *Vibrio fischeri* (also known as *Aliivibrio fischeri*) is a notable example of a bacterium that depends on multiple signaling systems to establish and maintain association with a host (*Miyashiro and Ruby, 2012*; *Verma and Miyashiro, 2013*; *Visick et al., 2021*). While a variety of marine animals serve as hosts for *V. fischeri*, the Hawaiian bobtail squid *Euprymna scolopes* is by far the best characterized, and this host–microbe association has emerged as a powerful system to model how signaling systems function in a natural host environment. From a specialized light organ located within the ventral side of the mantle, populations of *V. fischeri* emit bioluminescence that camouflage the host when viewed from below (*Jones and Nishiguchi, 2004*). Because *V. fischeri* grows on host-derived compounds within the light organ (*Graf and Ruby, 1998*; *Wasilko et al., 2019*), the association is considered a mutualistic symbiosis, in which each taxon benefits from their long-term and intimate interactions. The symbiosis is initially established after juvenile squid are exposed to seawater containing *V. fischeri* cells (*Lee and Ruby, 1994*), which enables bacterial mutants to be assessed for their ability to establish symbiosis, that is, to colonize, grow, and produce bioluminescence within the light organ.

The light-producing luciferase enzyme is encoded within the *lux* operon, which is transcribed when signaling by the LuxI/LuxR quorum-sensing system occurs (*Miyashiro and Ruby, 2012*; *Figure 1*).

Quorum sensing describes the phenomenon when bacteria synthesize, detect, and respond to small signaling molecules called autoinducers (*Whiteley et al., 2017*; *Papenfort and Bassler, 2016*). Mutants for either LuxI (autoinducer synthase) or LuxR (autoinducer receptor) fail to produce bioluminescence in vivo (*Visick et al., 2000*; *Yount et al., 2022*), which illustrates the significance of quorum sensing for the symbiosis to be established. In addition to the LuxI/LuxR system, two other quorum-sensing systems (AinS/AinR and LuxS/LuxPQ) affect bioluminescence production by indirectly regulating transcription of the *lux* operon (*Miyashiro and Ruby, 2012*; *Figure 1*). Under conditions of low autoinducer concentrations, either AinR or LuxPQ can trigger a phosphorelay that results in phosphorylation of the transcription factor LuxO (*Miyashiro et al., 2010*; *Kimbrough and Stabb, 2013*). In conjunction with the alternative sigma factor $\sigma^{54}$, LuxO activates the transcription of the small quorum regulatory RNA Qrr1 (*Miyashiro et al., 2010*; *Kimbrough and Stabb, 2015*), which lowers the ability of *V. fischeri* to enhance bioluminescence production (*Figure 1*). In contrast to the critical role that the LuxI/LuxR system has on establishing symbiosis, the impact of signaling by these other quorum-sensing systems is more nuanced, with knockout mutants for specific pathway components exhibiting symbiosis-related phenotypes that are observable only when introduced to juvenile squid as an inoculum mixed with another strain type (reviewed in *Verma and Miyashiro, 2013*). For instance, a Δ*qrr1* mutant can establish a light organ symbiosis with bacterial abundance and bioluminescence emission levels that are indistinguishable from squid colonized with the wild-type strain (*Miyashiro et al., 2010*). However, when juvenile squid are exposed to an inoculum evenly mixed with Δ*qrr1* mutant and wild-type strains, they later feature light organs containing threefold fewer Δ*qrr1* cells than wild-type cells (*Miyashiro et al., 2010*), which suggests that the expression of Qrr1 provides an advantage for *V. fischeri* to establish symbiosis when other potential founder cells are also present.

The primary structure of LuxO features an N-terminal regulatory domain, a central catalytic domain, and a C-terminal DNA-binding domain that define this transcription factor as a Group I bacterial enhancer binding protein (bEBP) (*Bush and Dixon, 2012*). As extensively reviewed elsewhere (*Bush and Dixon, 2012*; *Gao et al., 2020*), bEBPs bind upstream of $\sigma^{54}$-dependent promoters and hydrolyze ATP to induce the conformational changes within the RNA polymerase/$\sigma^{54}$/promoter complex that facilitate transcription initiation. Mechanistic studies in other *Vibrionaceae* have shown that the ATPase activity of LuxO is controlled by its N-terminal regulatory domain (*Boyaci et al., 2016*), which consists of a REC domain that participates in a phosphorelay (*Freeman and Bassler, 1999a*; *Figure 1*). In its unphosphorylated form, LuxO is inactive, with a 20-residue linker that connects the regulatory and catalytic domains occupying the active site within the catalytic domain to block nucleotide binding (*Boyaci et al., 2016*). The linker is a structural feature reportedly unique to LuxO, and its position within the active site is stabilized by hydrogen bonds with the regulatory and catalytic domains (*Boyaci et al., 2016*). This linker model is also supported for the LuxO homolog of *V. fischeri*—V114 is a residue within the regulatory domain that is predicted to interact with the linker region, and its substitution with either alanine or glycine results in a variant of LuxO with elevated activity (*Kimbrough and Stabb, 2015*). Phosphorylation of an aspartate conserved among REC domains (D55 in the LuxO homolog of *V. fischeri*) is predicted to displace the linker (*Freeman and Bassler, 1999b*), which enables activation of LuxO and transcriptional initiation of the *qrr1* promoter ($P_{qrr1}$). Phosphorylation of LuxO occurs when the histidine kinases that serve as quorum-sensing receptors are unbound with ligand (*Kimbrough and Stabb, 2013*; *Figure 1*); consequently, conditions of low cell density result in LuxO activity and transcriptional activation of $P_{qrr1}$ (*Miyashiro et al., 2010*). As the population grows, higher levels of the respective autoinducer ligands promote LuxO dephosphorylation, thereby lowering Qrr1 expression and permitting enhanced bioluminescence production (*Figure 1*).

Despite these advances in understanding the molecular mechanisms by which quorum-sensing systems regulate transcriptional activity of $P_{qrr1}$ in *V. fischeri* (*Miyashiro et al., 2010*; *Stabb and Visick, 2013*), how Qrr1 expression is controlled during symbiosis establishment remains unclear. For instance, prior to entering the light organ, bacterial cells are collected from the environment and form aggregates that are densely packed (*Visick et al., 2021*; *Nawroth et al., 2017*). A priori, such cellular arrangements are predicted to engage in quorum sensing and lower transcriptional activation of $P_{qrr1}$ (*Figure 1*), which would seemingly prevent cells from expressing Qrr1 to gain an advantage in host colonization. Here, we report a regulatory mechanism that enables *V. fischeri* to avoid this predicament. In particular, we reveal that the signaling pathways associated with aggregation and quorum sensing are connected in *V. fischeri*, and we demonstrate that this connection contributes to

host colonization. Genetic analysis shows that $\sigma^{54}$-dependent transcription of $P_{qrr1}$ can be activated by two distinct bEBPs that depend on overlapping *cis* regulatory elements, thereby resulting in a gene regulation module that resembles an OR logic gate, in which activation of either bEBP results in Qrr1 expression. Bioinformatic analysis suggests the potential for dual bEBP activation of Qrrs in approximately half of the other clades of the *Vibrionaceae* family, which suggests that this regulatory mechanism is widespread among biomedically and ecologically important taxa.

## Results

### BinK inhibits transcriptional activation of Qrr1

In *V. fischeri*, one of the autoinducers involved in quorum sensing is *N*-octanoyl homoserine lactone (C8 HSL), which is synthesized by AinS and detected by the histidine kinase AinR (*Kimbrough and Stabb, 2013*; *Gilson et al., 1995*; *Figure 1*). The phosphorelay that is initiated when AinR detects C8 HSL leads to lower transcriptional activity of $P_{qrr1}$ (*Kimbrough and Stabb, 2013*), which indicates that quorum sensing attenuates Qrr1 expression. Consistent with this model, the high cell density associated with colonies leads to low $P_{qrr1}$ transcriptional activity. Previously, we described a screen designed to identify genetic factors that inhibit $P_{qrr1}$ activity within colonies (*Miyashiro et al., 2014*). More specifically, the screen had been performed by introducing a GFP-based, transcriptional reporter for $P_{qrr1}$ ($P_{qrr1}$::*gfp*) into a Tn*5*-mutant library derived from wild-type strain ES114, selecting for conjugants by plating cells onto solid rich medium, and screening the resulting colonies for increased GFP fluorescence. One mutant resulting from the screen contains a transposon insertion within the gene *binK* (*VF_A0360*), which encodes the hybrid histidine kinase BinK (*Brooks and Mandel, 2016*; *Figure 2A*). To validate that the disruption of *binK* conferred increased $P_{qrr1}$ activity, we assessed the $P_{qrr1}$::*gfp* reporter in a Δ*binK* mutant that was previously reported (*Brooks and Mandel, 2016*). When grown on solid medium to high cell density, the Δ*binK* mutant exhibited 3.7-fold higher levels of GFP fluorescence relative to WT (*Figure 2B*), which suggests that $P_{qrr1}$ is transcriptionally active in cells lacking BinK. Wild-type levels of GFP fluorescence were observed in a Δ*binK* mutant expressing *binK in trans* (*Figure 2B*), demonstrating genetic complementation. Together, these data suggest that conditions of high cell density fail to lower Qrr1 expression in cells lacking BinK.

Our discovery that Qrr1 expression is controlled by BinK is of interest because this sensor kinase is known to affect how *V. fischeri* colonizes the light organ. BinK is part of a complex regulatory pathway that governs biofilm formation in *V. fischeri* by controlling the production of symbiosis polysaccharide (Syp) (*Figure 2C*), which is thought to comprise the major matrix component of the cellular aggregate that forms prior to *V. fischeri* entering the light organ. Syp production depends on transcriptional activation of an 18-gene *syp* locus by $\sigma^{54}$ and the bEBP SypG (*Yip et al., 2005*). The hybrid histidine kinase RscS initiates a phosphorelay that ultimately phosphorylates SypG, thereby activating the bEBP to promote $\sigma^{54}$-dependent transcription of the *syp* genes that are required for biofilm formation (*Yip et al., 2006*). BinK is hypothesized to inhibit biofilm formation by either directly or indirectly dephosphorylating SypG to lower transcriptional activation of the *syp* locus (*Brooks and Mandel, 2016*; *Ludvik et al., 2021*).

In *V. fischeri*, Qrr1 post-transcriptionally represses the expression of LitR, which is a transcription factor that enhances transcription of the *lux* operon, so cells expressing Qrr1 produce low levels of bioluminescence (*Miyashiro et al., 2010*). Our observation of increased $P_{qrr1}$ activity in the Δ*binK* mutant prompted us to investigate bioluminescence production throughout growth in culture. The Δ*binK* mutant produces wild-type levels of bioluminescence, including when the bioluminescence emission per cell unit (specific luminescence) amplifies during exponential growth (*Figure 2D*), which seemingly suggests that BinK has no impact on how quorum sensing regulates bioluminescence production. However, when originally assessed in a biofilm assay, the Δ*binK* mutant also phenocopied the wild-type strain unless the biofilm pathway was also induced, for example, by overexpressing the histidine kinase RscS (*Figure 2C*), which revealed that BinK inhibits biofilm formation (*Brooks and Mandel, 2016*). Therefore, we hypothesized that phenotypes associated with the Δ*binK* allele are similarly masked in bioluminescence assays. To test this hypothesis, we measured bioluminescence production of strains harboring the *rscS\** allele, which overexpresses RscS (*Yip et al., 2006*). Relative to the wild-type strain, the *rscS\** mutant exhibited a specific bioluminescence profile with a lower peak and less amplification (*Figure 2—figure supplement 1*). The specific bioluminescence profile of the

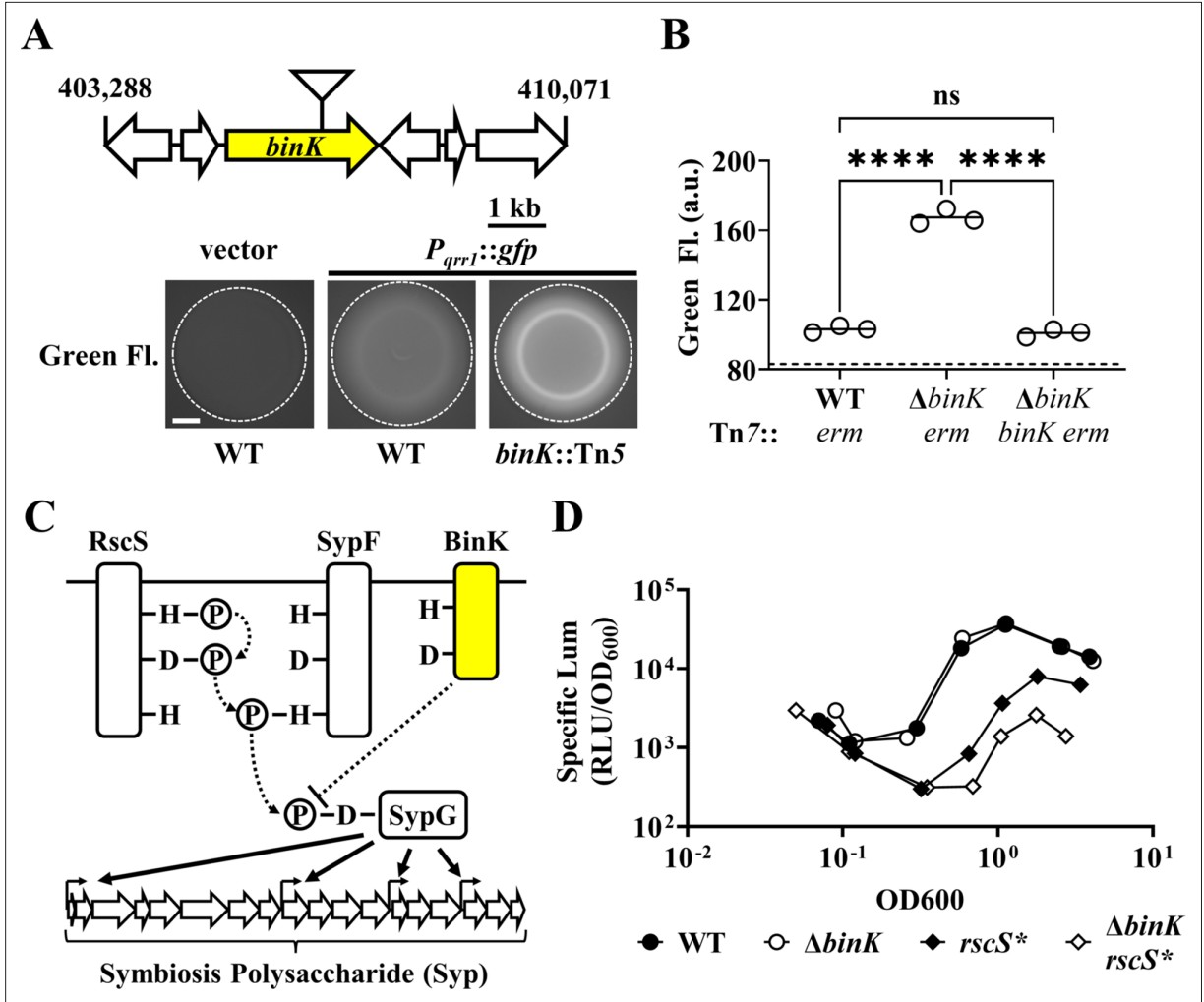

**Figure 2.** BinK inhibits the expression of the sRNA Qrr1. (**A**) *Top*, Tn*5* insertion location within *VF_A0360* (*binK*). *Below*, Green fluorescence images associated with ES114 (WT) and DRO22 (*binK*::Tn*5*) harboring pTM268 (P$_{qrr1}$::*gfp*) or pVSV105 (vector). Dotted circle indicates border of the spot of bacterial growth resulting from placing a cell suspension on the surface of solid rich medium and incubating the sample at 28°C for 24 hr. Scale bar = 1 mm. (**B**) Green fluorescence levels of TIM313 (WT Tn*7*::*erm*), MJM2481 (Δ*binK* Tn*7*::*erm*), and TIM412 (Δ*binK* Tn*7*::[*binK erm*]) harboring pTM268. Point = green fluorescence of a spot (*N* = 3), bar = group mean. Dotted line = autofluorescence cutoff. One-way analysis of variance (ANOVA; $F_{2,6}$ = 466.9, p < 0.0001); Tukey's post hoc test with p-values corrected for multiple comparisons (n.s. = not significant, ****p < 0.0001). (**C**) Signaling pathway for Syp-dependent biofilm formation in *V. fischeri* ES114. Phosphoryl groups are relayed (dotted arrows) from RscS to the HPT domain of SypF for phosphotransfer to SypG. SypG activates σ54-dependent transcription of the *syp* locus to promote biofilm formation. BinK negatively regulates this process and likely changes the phosphorylation of SypG (directly or indirectly). (**D**) *Top*, Bioluminescence assay of ES114 (WT), MJM2251 (Δ*binK*), MJM1198 (*rscS**), and MJM2255 (Δ*binK rscS**). Point = specific luminescence (RLU/OD$_{600}$) of indicated strain at the indicated turbidity (OD$_{600}$). Shown are points derived from a representative culture (*N* = 3). Experimental trials: 2.

The online version of this article includes the following source data and figure supplement(s) for figure 2:

**Source data 1.** Source data for *Figure 2B, D*.

**Figure supplement 1.** Analyses of bioluminescence assay described in *Figure 2D*.

Δ*binK rscS** mutant featured an even lower peak and lower amplification (*Figure 2—figure supplement 1*), which suggests that RscS overexpression reveals the ability of BinK to inhibit bioluminescence production. Taken together, these results provide evidence that the altered cellular physiology of Δ*binK* leads to attenuated bioluminescence production and lowered amplification under conditions of high cell density, which is consistent with elevated Qrr1 levels.

## Enhanced crypt colonization by the ΔbinK mutant is independent of Qrr1

Qrr1 and BinK are significant factors in the life cycle of *V. fischeri* because they each impact how *V. fischeri* cells initially establish symbiosis with *E. scolopes*. BinK inhibits the aggregation that occurs among environmental *V. fischeri* cells collected by the light organ, such that cells lacking BinK form large aggregates prior to light organ entry (*Brooks and Mandel, 2016*; *Ludvik et al., 2021*). In addition, animals exposed to an inoculum mixed evenly with a ΔbinK mutant and its wild-type parental strain result in approximately fourfold more ΔbinK cells than wild-type cells within their light organs (*Brooks and Mandel, 2016*), which suggests that BinK inhibits the ability of a cell to establish symbiosis in the context of other colonizing bacteria. In contrast, Qrr1 provides an advantage to *V. fischeri* when establishing symbiosis in the presence of other cells, as squid exposed to an inoculum mixed evenly with a Δqrr1 mutant and its wild-type parental strain lead to threefold fewer Δqrr1 cells than wild-type cells within colonized animals (*Miyashiro et al., 2010*). Consequently, the discovery that $P_{qrr1}$ expression is elevated within a ΔbinK mutant led us to investigate whether this regulatory connection impacts how *V. fischeri* establishes symbiosis, particularly in the context of competition.

Upon symbiosis establishment, the light organ contains up to six independent populations of *V. fischeri*, with each population housed within an epithelium-lined crypt space (*Montgomery and McFall-Ngai, 1993*). Because the isolation of colony-forming units (CFUs) requires tissue homogenization, approaches based on counting CFUs to quantify cellular abundance in vivo inherently disrupt the location of the strains within the light organ, thereby precluding insight that can be deduced from this knowledge. For example, identification of a strain being present within a colonized crypt space reveals that the strain initially accessed the crypt and grew. Using this approach, we first determined where the Δqrr1 mutant and a wild-type competitor strain reside within the light organ by differentially labeling each strain type with fluorescent proteins and assessing their location within host tissue by fluorescence microscopy (*Verma and Miyashiro, 2016*; *Figure 3A*). As expected, most light organs contained populations in several crypt spaces (*Figure 3B, C*), which indicated that multiple colonization events had occurred within each animal. Most colonized crypt spaces contained only one strain type (*Figure 3B*), which is consistent with populations arising from only one to two cells that enter the corresponding crypt spaces (*Wollenberg and Ruby, 2009*). Few crypt spaces harbored the Δqrr1 mutant (*Figure 3B, C*), which suggests that the majority of populations were founded by wild-type cells. In contrast, when the inoculum contained an equal mix of differentially labeled wild-type cells, no difference was observed in the number of crypt spaces colonized by YFP- or CFP-labeled strains (*Figure 3B, C*). Consequently, these results suggest that the competitive defect of the Δqrr1 mutant reported previously is due to fewer crypt spaces being initially accessed by the mutant.

We next used this microscopy-based approach to investigate the ΔbinK mutant. Following co-inoculation with the wild-type competitor strain, the ΔbinK mutant occupied most of the crypt spaces (*Figure 3D*), which suggests that ΔbinK cells founded more populations than competitor cells and explains the previous observation of higher relative abundance of ΔbinK cells in squid co-inoculated with those strain types (*Brooks and Mandel, 2016*). To determine whether Qrr1 impacts this effect, we also examined light organs of animals exposed to an inoculum mixed evenly with ΔbinK Δqrr1 mutant and the wild-type competitor. The ΔbinK Δqrr1 mutant occupied a minority of crypt spaces (*Figure 3E*), which suggests that Qrr1 also promotes the ability of the ΔbinK mutant to access crypt spaces. Because the ΔbinK mutant forms large aggregates, we also considered whether Qrr1 affects this process by determining the extent to which the ΔbinKΔqrr1 mutant could form aggregates. As expected, ΔbinK formed larger aggregates than WT cells (*Figure 3F*), which highlights the inhibitory role of BinK on aggregation formation that was previously reported (*Brooks and Mandel, 2016*). Most of the aggregates formed by the ΔbinKΔqrr1 mutant were also large (*Figure 3F*), which suggests that the impact of Qrr1 on aggregation formation is minimal. Furthermore, when juvenile squid were exposed to an inoculum containing the ΔbinK and ΔbinK Δqrr1 mutants, far more crypts contained the ΔbinK mutant than the double mutant (*Figure 3—figure supplement 1*), which suggests the enhanced aggregation of cells with the ΔbinK allele does not mitigate the impact of Qrr1 during crypt colonization. Taken together, these data suggest that the Δqrr1 allele is epistatic to the ΔbinK allele during symbiosis establishment, which provides evidence that Qrr1 affects the ability of *V. fischeri* to enter a crypt space after the aggregation phase.

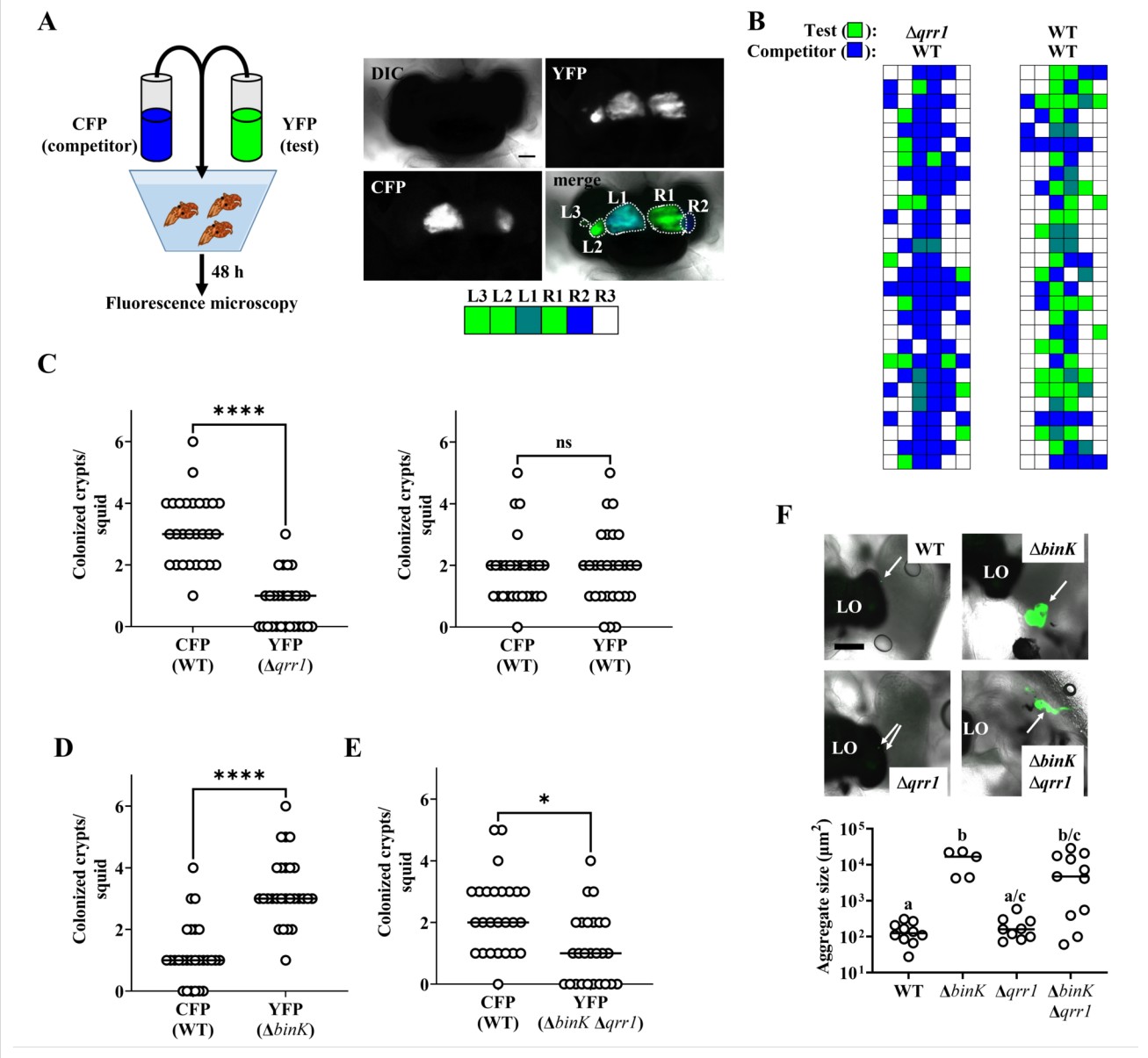

**Figure 3.** Qrr1 enhances the ability of *V. fischeri* to access crypt spaces. (**A**) *Left*, experimental design of squid co-inoculation assays with YFP-labeled test strain and CFP-labeled wild-type competitor strain. *Right*, example image montage illustrating a light organ featuring populations comprised of cells expressing YFP or CFP. Dotted line = boundary of an individual population. Scale bar = 100 μm. Row of boxes below the image indicate the strain type(s) present within each predicted crypt space of the light organ; blue = CFP+ YFP−, green = YFP+ CFP−, hatched = CFP+ YFP+, white = CFP− YFP−. For panels B–E, experimental trials = 2. (**B**) *Left*, TIM305 (Δ*qrr1*) as test strain. *Right*, ES114 (WT) as test strain. Each row represents an individual animal (*N* = 28). (**C**) Number of crypts colonized by indicated strains per squid in panel B. Wilcoxon test (****p < 0.0001, n.s. = not significant). (**D**) Δ*binK*. Number of crypts colonized by MJM2251 (Δ*binK*) as test strain. *N* = 27. Wilcoxon test (****p < 0.0001). (**E**) Δ*binK* Δ*qrr1*. Number of crypts colonized by EDR010 (Δ*binK* Δ*qrr1*) as test strain. *N* = 26. Wilcoxon test (*p < 0.05). (**F**) Aggregation assay with ES114 (WT), MJM2251 (Δ*binK*), and EDR010 (Δ*binK* Δ*qrr1*) labeled with YFP. *Top*, merged brightfield and yellow fluorescence (green) images of aggregates (arrows) formed by indicated strains. LO = light organ. Scale bar = 200 μm. *Bottom*, quantification of aggregate size. Kruskal–Wallis (*H* = 16.79, *d.f.* = 3, p = 0.0008); Dunn's post hoc test with p-values corrected for multiple comparisons (same letter = not significant, a/c and b/c = p < 0.05, a/b = p < 0.01). Experimental trials: 2.

The online version of this article includes the following source data and figure supplement(s) for figure 3:

**Source data 1.** Source data for *Figure 3B–F*.

**Figure supplement 1.** Impact of Qrr1 on crypt colonization is independent of BinK.

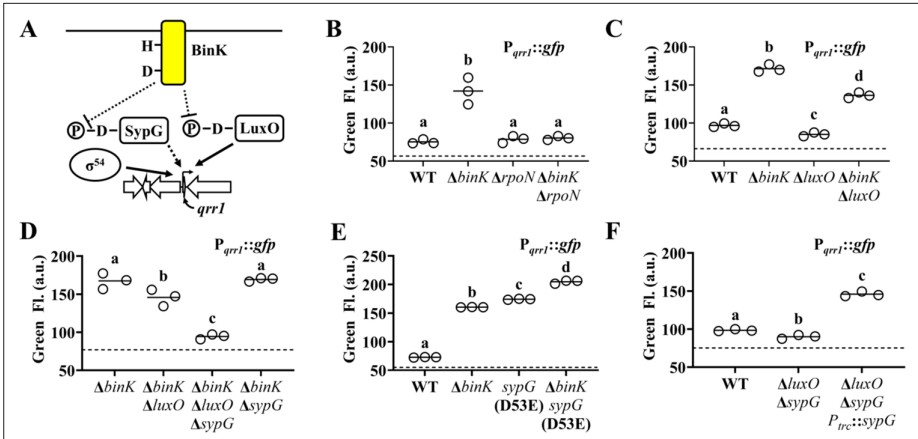

**Figure 4.** SypG activates σ⁵⁴-dependent transcription of *qrr1*. (**A**) Proposed model of BinK-dependent regulation of Qrr1 expression. (**B**) Green fluorescence levels of ES114 (WT), MJM2251 (Δ*binK*), KRG004 (Δ*rpoN*), and KRG011 (Δ*binK* Δ*rpoN*) harboring pTM268 (P*qrr1*::*gfp*). Dotted line = autofluorescence cutoff. One-way analysis of variance (ANOVA; $F_{3,8}$ = 35.69, p < 0.0001); Tukey's post hoc test with p-values corrected for multiple comparisons (same letter = not significant, different letters = p < 0.001). (**C**) Green fluorescence levels of ES114 (WT), MJM2251 (Δ*binK*), TIM306 (Δ*luxO*), and (Δ*binK* Δ*luxO*) harboring pTM268 (P*qrr1*::*gfp*). Dotted line = autofluorescence cutoff. One-way ANOVA ($F_{3,8}$ = 367.4, p < 0.0001). (**D**) Green fluorescence levels of MJM2251 (Δ*binK*), EDR009 (Δ*binK* Δ*luxO*), EDR014 (Δ*binK* Δ*sypG*), and EDR013 (Δ*binK* Δ*luxO* Δ*sypG*) harboring pTM268 (P*qrr1*::*gfp*). Dotted line = autofluorescence cutoff. One-way ANOVA ($F_{3,8}$ = 60.66, p < 0.0001). (**E**) Green fluorescence levels of ES114 (WT), MJM2251 (Δ*binK*), MJM4982 [*sypG*(D53E)], and MJM4983 [Δ*binK sypG*(D53E)] harboring pTM268 (P*qrr1*::*gfp*). Dotted line = autofluorescence cutoff. One-way ANOVA ($F_{3,8}$ = 3921, p < 0.0001). (**F**) Green fluorescence levels of TIM313 (WT), EDS008 (Δ*luxO* Δ*sypG*), and EDS010 (Δ*luxO* Δ*sypG* P*trc*::*sypG*) harboring pEDR003 (P*qrr1*::*gfp*) and grown on 150 µM IPTG. Dotted line = autofluorescence cutoff. One-way ANOVA ($F_{2,6}$ = 438.8, p < 0.0001).

The online version of this article includes the following source data and figure supplement(s) for figure 4:

**Source data 1.** Source data for *Figure 4B–F*.

**Figure supplement 1.** Comparisons of LuxO with other Class I bacterial enhancer binding proteins (bEBPs) encoded by *V. fischeri*.

**Figure supplement 2.** Alignment of SypG with LuxO crystal structure.

**Figure supplement 3.** LuxO does not activate promoters of the *syp* locus in *V. fischeri*.

## The bEBP SypG activates σ⁵⁴-dependent transcription of P*qrr1* in *V. fischeri*

To determine how P*qrr1* is activated in the Δ*binK* mutant, we considered factors known to promote transcription of *qrr1*. As with the *qrr* genes in other *Vibrionaceae* members (**Lenz et al., 2004**), the promoter region of *qrr1* in *V. fischeri* (**Figure 4A**) features nucleotides corresponding to the canonical −24 and −12 sites (TGGCA-N7-TGC) that facilitate binding by the alternative sigma factor σ⁵⁴ (**Bush and Dixon, 2012**). To test whether the P*qrr1* activity observed in the Δ*binK* mutant depends on σ⁵⁴, we knocked out the *rpoN* gene that encodes σ⁵⁴ from the Δ*binK* mutant and assessed P*qrr1*::*gfp* activity in the resulting Δ*rpoN* Δ*binK* double mutant. GFP levels in the double mutant were attenuated and comparable to the low levels of the Δ*rpoN* single mutant (**Figure 4B**), which indicates that the activity of P*qrr1* of Δ*binK* cells depends on σ⁵⁴.

Transcriptional activation of σ⁵⁴-dependent promoters critically depends on a bEBP interacting with nucleotides upstream of the promoter and hydrolyzing ATP to induce the conformation changes in the σ⁵⁴-RNA polymerase–promoter complex that facilitate transcriptional activation (**Bush and Dixon, 2012**). Therefore, we next considered whether the P*qrr1* activity observed in the Δ*binK* mutant depends on LuxO, which is the only bEBP known to activate σ⁵⁴-dependent transcription of P*qrr1* (**Miyashiro et al., 2010**). While the GFP fluorescence level of a Δ*luxO* Δ*binK* mutant was lower than that of the Δ*binK* mutant (**Figure 4C**), it was consistently higher than that of the wild-type strain, suggesting that LuxO is only partially responsible for σ⁵⁴-dependent P*qrr1* activity in the Δ*binK* mutant.

The partial effect of LuxO described above suggests that a different bEBP also facilitates the σ$^{54}$-dependent P$_{qrr1}$ activity observed in the Δ*binK* mutant. In addition to LuxO, the genome of ES114 encodes five other class I bEBPs: SypG, NtrC, VF_1401, FlrC, and VpsR. Of these other bEBPs, SypG stood out as a candidate for LuxO-independent activation of P$_{qrr1}$ for three reasons: (1) SypG-dependent gene expression is elevated in the Δ*binK* mutant (*Brooks and Mandel, 2016*), (2) the primary structure of SypG is most identical to that of LuxO (*Figure 4—figure supplement 1*) and predicted to form many of the structural features underlying LuxO function (*Boyaci et al., 2016*; *Figure 4—figure supplement 2*), and (3) WT cells harboring a multi-copy plasmid containing *sypG* exhibit elevated P$_{qrr1}$ activity (*Miyashiro et al., 2014*). To test whether SypG affects the LuxO-independent P$_{qrr1}$ activity of Δ*binK* mutant cells, we constructed a Δ*binK* Δ*luxO* Δ*sypG* triple mutant. GFP fluorescence was lower in the triple mutant relative to the Δ*binK* Δ*luxO* mutant (*Figure 4D*), which suggests that SypG promotes LuxO-independent P$_{qrr1}$ activity in cells lacking *binK*. Notably, P$_{qrr1}$ activity remained high in a Δ*binK* Δ*sypG* double mutant (*Figure 4D*), which suggests that LuxO is the primary activator of P$_{qrr1}$ in the Δ*binK* mutant.

Previous studies have shown that transcriptional expression of the *syp* locus depends on SypG and is elevated in the Δ*binK* mutant (*Brooks and Mandel, 2016*; *Ludvik et al., 2021*; *Hussa et al., 2008*). To determine whether the increased LuxO activity associated with the Δ*binK* mutant also contributes to *syp* expression, we assessed transcriptional activity of the promoters for *sypA* (P$_{sypA}$) and *sypP* (P$_{sypP}$). Both promoters show elevated activity in the Δ*binK* and Δ*binK* Δ*luxO* mutants but background levels in the Δ*binK* Δ*sypG* mutant (*Figure 4—figure supplement 3*), which is consistent with their expression in the Δ*binK* background depending on SypG but not LuxO. Using a mutant that expresses the phosphomimetic variant LuxO(D55E), we also found that the transcriptional activities of P$_{sypA}$ and P$_{sypP}$ remain inactive in cells with elevated LuxO activity (*Figure 4—figure supplement 3*), which suggests that phosphorylated LuxO does not promote transcription of the *syp* locus. Taken together, these results suggest that while the *syp* genes are insulated from LuxO, *qrr1* can be activated by both SypG and LuxO.

Like LuxO, SypG depends on phosphorylation of a conserved aspartate within its N-terminal REC domain for activation (*Hussa et al., 2008*). To determine whether activation of SypG increases P$_{qrr1}$ transcription, we utilized a *sypG*(D53E) allele, which encodes a phosphomimetic variant of SypG that promotes *syp* expression (*Ludvik et al., 2021*; *Hussa et al., 2008*). Cells encoding this active SypG variant express high P$_{qrr1}$ transcriptional activity (*Figure 4E*), which suggests that phosphorylated SypG leads to Qrr1 expression. A Δ*binK* *sypG*(D53E) mutant showed higher levels of P$_{qrr1}$ activity than either of the corresponding single mutants (*Figure 4E*), which suggests that BinK inhibits activation of factors other than SypG (e.g., LuxO). To determine whether wild-type SypG can also activate P$_{qrr1}$ in the presence of BinK, we evaluated P$_{qrr1}$ activity in response to SypG expression in cells that encode BinK. Using a Δ*luxO* Δ*sypG* mutant to eliminate background signal to P$_{qrr1}$ activity, we found that induction of *sypG* expression was sufficient to activate P$_{qrr1}$ transcription (*Figure 4F*). Taken together, we conclude that SypG is a bEBP that activates P$_{qrr1}$ in addition to the *syp* locus in *V. fischeri*.

## Quorum sensing does not inhibit SypG-dependent activation of *qrr1*

Based on our finding that SypG activates transcription of P$_{qrr1}$, we hypothesized that conditions that promote SypG activity would elevate the expression of Qrr1, which is significant because Qrr1-dependent regulation could occur under conditions of high cell density. To test this hypothesis, we first examined P$_{qrr1}$ activity in cells overexpressing RscS, which stimulates the expression of SypG-dependent genes (*Hussa et al., 2008*; *Figure 2C*). Using a plasmid containing the *rscS** allele described above, RscS was overexpressed in *V. fischeri* strains engineered to encode a P$_{qrr1}$::*gfp* reporter within its chromosome. When cell suspensions were spotted onto solid medium and incubated, the resulting surface structures featured pronounced heterogenous ridges (*Figure 5A*), which comprise the wrinkled-colony phenotype that depends on expression of the *syp* locus (*Yip et al., 2006*). Green fluorescence was observed throughout the structure (*Figure 5A*), particularly within the ridges, which suggests that P$_{qrr1}$ was activated from overexpressing RscS. In contrast, overexpression of RscS in a Δ*sypG* mutant resulted in smooth surface structures (*Figure 5A*), which indicates the wrinkled-colony phenotype depends on a functional SypG, as previously reported (*Hussa et al., 2008*). Furthermore, low green fluorescence was observed for the Δ*sypG* mutant (*Figure 5A*), which indicates low P$_{qrr1}$ activity and suggests that SypG activation by RscS overexpression results in Qrr1

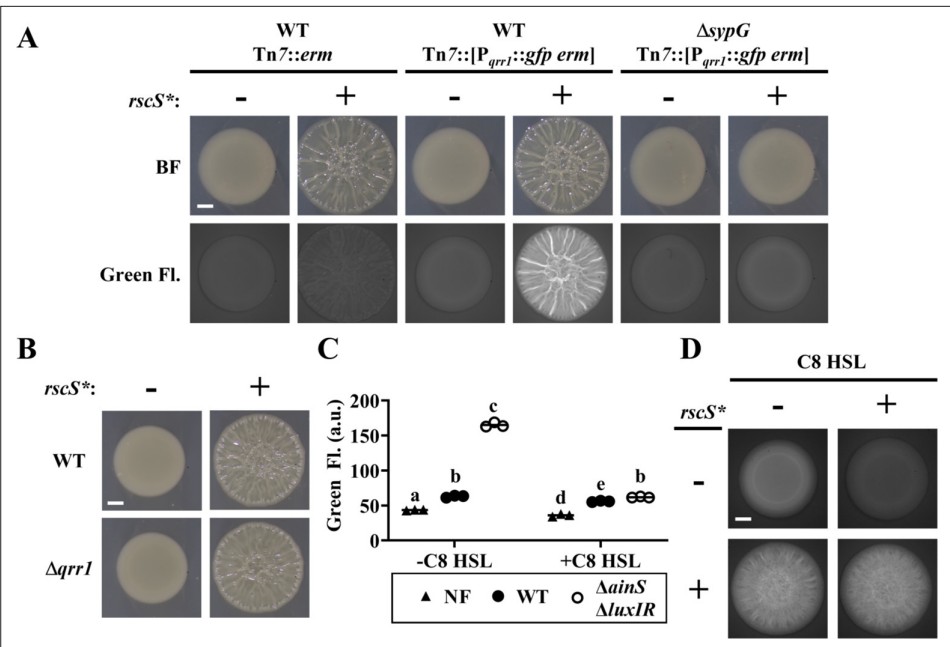

**Figure 5.** SypG activity overrides inhibition of P$_{qrr1}$ activity by quorum sensing. (**A**) Brightfield (top) and green fluorescence (bottom) images of representative spots of growth ($N = 3$) containing TIM313 (WT Tn7::*erm*), TIM303 (WT Tn7::[P$_{qrr1}$::*gfp erm*]), or EDS015 ($\Delta sypG$ Tn7::[P$_{qrr1}$::*gfp erm*]) harboring plasmid pKV69 (*rscS** = −) or pKG11 (*rscS** = +). Scale bar = 1 mm. (**B**) Brightfield images of representative spots of growth ($N = 3$) containing TIM303 (WT) or SSC005 ($\Delta qrr1$) harboring plasmid pKV69 (*rscS** = −) or pKG11 (*rscS** = +). Scale bar = 1 mm. (**C**) Green fluorescence of ES114 (WT) and JHK007 ($\Delta ainS \Delta luxIR$) harboring P$_{qrr1}$::*gfp* reporter pTM268 (circles) and grown ± 100 nM C8 HSL. ES114 harboring pVSV105 was used as a non-fluorescent control (NF). Two-way analysis of variance (ANOVA) revealed statistical significance for strain ($F_{2,12} = 2809$, p < 0.0001), C8 treatment ($F_{1,12} = 2233$, p < 0.0001), and their interaction ($F_{2,12} = 1480$, p < 0.0001); Tukey's post hoc test with p-values corrected for multiple comparisons (same letter = not significant, b/e = p < 0.05, a/d = p < 0.01, other combinations of different letters = p < 0.0001). (**D**) Green fluorescence images of representative spots of growth ($N = 3$) containing KRG016 ($\Delta ainS$ $\Delta luxIR$ Tn7::P$_{qrr1}$::*gfp*) harboring plasmid pKG11 (*rscS** = +) or pKV69 (*rscS** = −) on medium ± 100 nM C8 HSL. Scale bar = 1 mm.

The online version of this article includes the following source data for figure 5:

**Source data 1.** Source data for **Figure 5C**.

expression. However, a $\Delta qrr1$ mutant formed wrinkled colonies in response to overexpression of RscS (**Figure 5B**), which suggests that Qrr1 does not promote the process of wrinkled-colony formation.

We also investigated whether quorum sensing impacts SypG-dependent activation of P$_{qrr1}$. In *V. fischeri*, signaling by the histidine kinase AinR in response to C8 HSL autoinducer results in lowered P$_{qrr1}$ activity (**Figure 1** and **Kimbrough and Stabb, 2013**). The low P$_{qrr1}$ activity observed in the spots of growth (**Figure 2A**) suggests that the level of C8 HSL is already elevated within the high cell density conditions, which would prevent our ability to detect a response to additional C8 HSL. Therefore, we introduced the P$_{qrr1}$::*gfp* reporter into the chromosome of the $\Delta ainS$ mutant JHK007 (**Kimbrough and Stabb, 2013**), which does not produce the C8 HSL synthase AinS (**Gilson et al., 1995**). JHK007 also contains deletions of *luxI* and *luxR*, which contribute to an unknown mechanism that inhibits activation of P$_{qrr1}$ through AinR signaling (**Kimbrough and Stabb, 2013**). Consistent with this previous report, JHK007 showed elevated GFP fluorescence, which suggests high P$_{qrr1}$ activity in the absence of HSL-based autoinducers (**Figure 5C**). Supplementing media with C8 HSL was sufficient to lower GFP fluorescence (**Figure 5C**), which indicates that C8 HSL inhibits P$_{qrr1}$ activity. Using this experimental setup, we next assessed whether increased SypG activity could interfere with the ability of C8 HSL to inhibit P$_{qrr1}$ activity through the introduction of a plasmid harboring the *rscS** allele. As expected, overexpression of RscS resulted in wrinkled colonies with elevated GFP fluorescence (**Figure 5D**). However, the presence of C8 HSL did not alter the wrinkling phenotype and failed to lower GFP fluorescence, which suggests that P$_{qrr1}$ activity remained elevated in those spots. Taken together, these results suggest that

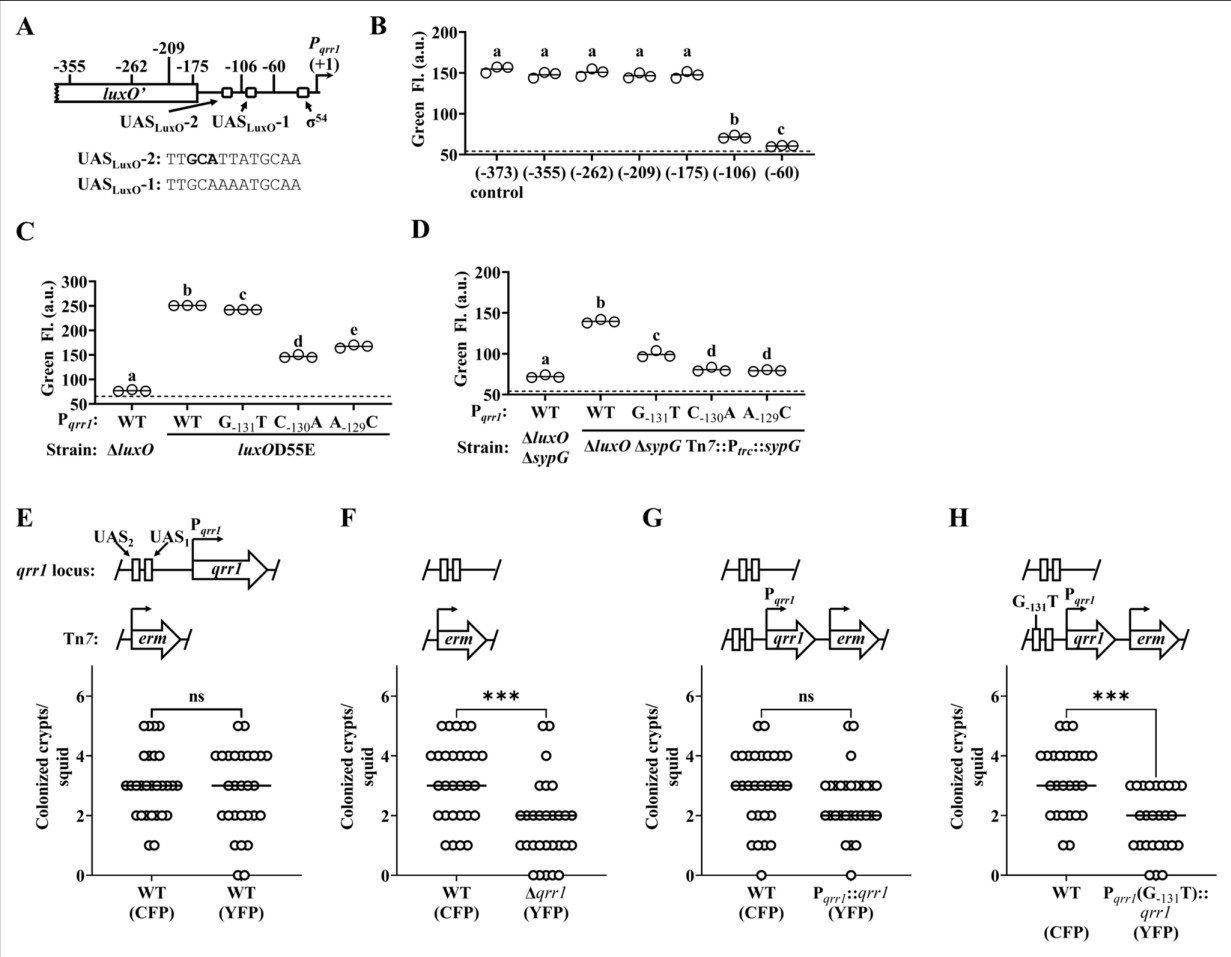

**Figure 6.** SypG activates σ54-dependent transcription of *qrr1*. (**A**) Elements within P$_{qrr1}$ that facilitate σ54-dependent transcriptional activation. Sequence corresponds to 13 bp UAS$_{LuxO}$-2, with nucleotides (−131)–(−129) that were individually mutated by site-directed mutagenesis shown in bold. (**B**) Green fluorescence levels of EDS010 (ΔluxO ΔsypG Tn7::[P$_{trc}$::sypG erm]) harboring P$_{qrr1}$::*gfp* reporter plasmids pEDR003, pEDR011 (−355), pEDR010 (−262), pEDR012 (−209), pEDR006 (−175), pEDR009 (−106), and pEDR008 (−60). Dotted line = autofluorescence cutoff. One-way analysis of variance (ANOVA; $F_{6,14}$ = 411.7, p < 0.0001). (**C**) Green fluorescence levels of TIM306 (ΔluxO) and CL59 (luxOD55E) harboring P$_{qrr1}$::*gfp* reporter plasmids pEDR003, pEDS004 (G$_{−131}$T), pEDS005 (C$_{−130}$A), and pEDS006 (A$_{−129}$C). The luxOD55E encodes a variant of LuxO that exhibits high activity in colonies. Dotted line = autofluorescence cutoff. One-way ANOVA ($F_{4,10}$ = 2998, p < 0.0001). (**D**) Green fluorescence levels of EDS008 (ΔluxO ΔsypG) and EDS010 (ΔluxO ΔsypG Tn7::P$_{trc}$::sypG) harboring P$_{qrr1}$::*gfp* reporter plasmids pEDR003 (WT), pEDS004 (G$_{−131}$T), pEDS005 (C$_{−130}$A), and pEDS006 (A$_{−129}$C). Dotted line = autofluorescence cutoff. One-way ANOVA ($F_{4,10}$ = 712.5, p < 0.0001). (**E–H**) Squid colonization assays, with each graph showing the number of crypts colonized by CFP-labeled TIM313 (WT) and the indicated YFP-labeled test strain. The diagram above each graph illustrates the genetic composition of the *qrr1* locus (top) and Tn7 integration site (bottom) in the corresponding test strain. Test strains are TIM313 (WT), KRG021 (Δqrr1), KRG018 (P$_{qrr1}$::qrr1), and (**D**) KRG019 (P$_{qrr1}$ (G$_{−131}$T)::qrr1). Wilcoxon test (***p < 0.001, n.s. = not significant).

The online version of this article includes the following source data for figure 6:

**Source data 1.** Source data for *Figure 6B–H*.

SypG-dependent activation of P$_{qrr1}$ is insensitive to autoinducer and furthermore indicate a mechanism by which *V. fischeri* can express Qrr1 even when cells conduct quorum sensing.

## Role for SypG-dependent regulation of Qrr1 during host colonization

We next asked whether the ability of SypG to activate Qrr1 expression affects light organ colonization. Because SypG also activates the *syp* gene expression (*Hussa et al., 2007*), we could not use strains containing the ΔsypG allele to address this question because such mutants would exhibit colonization defects due to failed aggregate formation. Therefore, we instead examined P$_{qrr1}$ for *cis* regulatory elements that could be mutated to specifically interfere with SypG-dependent activation of Qrr1 expression. To determine which regions upstream of P$_{qrr1}$ are necessary for SypG-dependent

regulation, we generated a set of P$_{qrr1}$::*gfp* reporter constructs of various lengths at the 5'-end (*Figure 6A*) and then evaluated P$_{qrr1}$ activity in the Δ*luxO* Δ*sypG* mutant with *sypG* expressed in trans (*Figure 4F*). From these measurements, we found two regions that contribute to SypG-dependent activation, one between −175 and −106 bp and a second between −106 and −60 bp (*Figure 6B*). Within each region, we identified a sequence similar to the TT**CT**CANNNTGMDWN motif previously reported as the UAS of SypG (UAS$_{SypG}$) (*Ray et al., 2013*), with nucleotides in bold being the only mismatches. Notably, each site also features perfect matches of the 13 bp motif TTGCAWWWTGCAA that corresponds to the UAS of LuxO (UAS$_{LuxO}$) reported in other *Vibrionaceae* members (*Lenz et al., 2004*; *Svenningsen et al., 2008*; *Figure 6A*), which raises the possibility that SypG and LuxO have overlapping UAS within each region.

The possibility of overlapping UAS$_{SypG}$ and UAS$_{LuxO}$ within P$_{qrr1}$ could complicate the strategy to target the UAS$_{SypG}$ within P$_{qrr1}$ to disrupt SypG-dependent regulation by also affecting how LuxO regulates Qrr1 expression. Therefore, we evaluated several nucleotides to determine their corresponding roles on the regulation of P$_{qrr1}$ by LuxO and SypG. Substitution of either cytosine or adenine within the first half of UAS$_{LuxO-2}$ (C$_{−130}$A and A$_{−129}$C, respectively) attenuated both LuxO- and SypG-dependent regulation of P$_{qrr1}$ activity (*Figure 6C, D*), which suggests that these nucleotides are important for both bEBPs to promote Qrr1 expression. In contrast, substitution of the guanine (G$_{−131}$T) had little impact on LuxO-dependent expression of P$_{qrr1}$ (*Figure 6C*) but decreased SypG-dependent expression (*Figure 6D*), which suggests this nucleotide plays a role in specifically mediating how SypG interacts with P$_{qrr1}$. While the latter result is surprising because the G$_{−131}$T substitution leads to this site more closely resembling the UAS$_{SypG}$ motif described above, it provided a means to disrupt SypG-dependent regulation of Qrr1 expression without affecting regulation by LuxO.

To determine whether regulation of Qrr1 by SypG affects how *V. fischeri* colonizes the light organ, we conducted a series of squid colonization assays using inoculums evenly mixed with a wild-type competitor strain and various test strains described below. Consistent with the data in *Figure 3C* that implicates Qrr1 as an important factor that promotes crypt access, a wild-type test strain colonized a comparable number of crypts as the competitor, but the Δ*qrr1* mutant colonized fewer crypts than the competitor (*Figure 6E, F*). Integration of a cassette including *qrr1* with its native promoter into the Δ*qrr1* mutant restored rates of crypt colonization comparable to the competitor strain (*Figure 6G*). However, a mutant containing the single SypG-relevant G$_{−131}$T substitution upstream of P$_{qrr1}$ resulted in fewer crypts being colonized (*Figure 6H*), which suggests that the ability of SypG to activate expression of Qrr1 is important for *V. fischeri* to access crypt spaces when competitor symbionts are present.

## Diversity of SypG-dependent activation of P$_{qrr1}$ among *Vibrionaceae*

*V. fischeri* is a member of the Fischeri clade of *Vibrionaceae*, which includes five species that reside in seawater habitats as well as within squid and fish (*Sawabe et al., 2013*). The genomes of *Vibrionaceae* members commonly feature two chromosomes of unequal size, with the larger chromosome referred to as Chromosome 1 (*Okada et al., 2005*). The genomes of representative Fischeri taxa encode homologs of Qrr1 and LuxO on Chromosome 1 (*Figure 7—figure supplement 1* and *Table 1*) and SypG on Chromosome 2 (*Table 1*). Gene synteny associated with each locus across taxa suggests that the genes encoding Qrr1 and the bEBPs were passed vertically within the Fischeri

**Table 1.** LuxO and SypG homologs in Fischeri clade.

| Taxon | Strain | LuxO homolog | Accession | SypG homolog | Accession | Identity | Similarity |
|---|---|---|---|---|---|---|---|
| *V. fischeri* | ES114 | WP_011261589.1 | NC_006840.2 | WP_011263835.1 | NC_006841.2 | 243/502 (48.41%) | 326/502 (64.94%) |
| *A. salmonicida* | LFI1238 | WP_173362130.1 | NC_011312.1 | WP_044583634.1 | NC_011313.1 | 236/504 (46.83%) | 320/504 (63.49%) |
| *A. sifiae* | NBRC 105001 | WP_172794763.1 | NZ_MSCP01000001.1 | WP_105064188.1 | NZ_MSCP01000002.1 | 238/495 (48.08%) | 316/495 (63.84%) |
| *A. wodanis* | AWOD1 | CED71013.1 | LN554846.1 | CED57805.1 | LN554847.1 | 238/494 (48.18%) | 321/494 (64.98%) |
| *A. logei* | 1S159 | WP_175365415.1 | NZ_MAJU01000008.1 | WP_065611272.1 | NZ_MAJU01000009.1 | 238/504 (47.22%) | 319/504 (63.29%) |

lineage (*Figure 7—figure supplement 2A, B*). For each taxon, alignment of the primary structures for the LuxO and SypG homologs revealed approximately 48% identity (*Table 1* and *Figure 7—figure supplement 3*). Among the five taxa, 44.4% (214/481) of residue positions in LuxO were identical to the corresponding SypG homolog (*Figure 7A, B* and *Figure 7—figure supplement 4*), which suggests that the functions associated with various domains of LuxO, including the regulatory linker and HTH domains, are also highly conserved in SypG. Together, these analyses based on bioinformatics suggest the possibility that SypG-dependent expression of Qrr sRNAs is conserved throughout the Fischeri clade.

We expanded our analysis to consider the *Vibrionaceae* family, which features species that are important in a variety of marine ecosystems, with many members able to cause disease in humans and other animals (*Grimes, 2020*). Reconstruction of the evolutionary history of the *Vibrionaceae* family has resulted in 22 distinct clades, including Fischeri (*Sawabe et al., 2013*). All clades except Rumioensis feature taxa encoding a LuxO homolog (*Table 2*), with the corresponding *luxO* gene located on Chromosome 1 in the 17 taxa for which fully assembled genomes are available. Each of the remaining clades represented by the indicated taxa for which only contigs are available also featured a *luxO* gene, and gene synteny analysis of the neighboring genes suggests an arrangement consistent with its location on Chromosome 1 (*Figure 7—figure supplement 5*). In addition, a putative Qrr is also encoded in opposite orientation of *luxO* in 20 of the 21 representative taxa that encode a LuxO homolog (*Figure 7—figure supplement 6*), which suggests that the LuxO–Qrr regulatory system is highly conserved among *Vibrionaceae* members.

Among the 21 taxa that encode a LuxO homolog, 12 of them also encode a SypG homolog (*Table 2*), and the corresponding *sypG* gene resides within a gene cluster that resembles the *syp* locus of *V. fischeri*. However, in contrast to the Fischeri clade, most taxa of other SypG-positive clades within the *Vibrionaceae* that have complete genomes encode the *syp* locus on Chromosome 1 (*Table 2*), which suggests the possibility that the *syp* locus was acquired by a progenitor of the Fischeri clade that arose after diversification from other *Vibrionaceae* lineages. Despite this possibility of independent acquisition events, the SypG homologs encoded by non-Fischeri taxa also exhibit high amino acid sequence identity to the corresponding LuxO homologues (*Table 2*), including the same structural features involved in regulating activity (*Figure 7C*). To gain insight into the evolutionary history associated with the SypG homolog encoded by the Chromosome I of these other taxa, we evaluated the genomic context of *pepN*, which is genetically linked to the *syp* locus in several taxa but also highly conserved among all *Vibrionaceae*. Gene synteny analysis of *pepN* suggests that genome rearrangement likely contributed to certain taxa losing the *syp* locus, and consequently *sypG* (*Figure 7—figure supplement 7*). Taken together, these observations suggest that while the *Vibrionaceae* lineage has undergone significant diversification with SypG, those taxa that encode both SypG and LuxO have the potential for SypG-dependent activation of Qrr sRNAs.

Finally, to test the possibility of SypG-dependent activation of $P_{qrr1}$ in taxa other than *V. fischeri*, we considered the fish pathogen *Aliivibrio salmonicida* strain LFI1238, which encodes a SypG homolog (SypG$_{As}$) with nearly 47% identity to its LuxO homolog (*Table 1* and *Figure 7—figure supplement 3*). Similar to *V. fischeri*, the genome of LFI1238 also features a single *qrr* gene (*qrr1$_{As}$*) with a promoter region ($P_{qrr1AS}$) that contains motifs associated with $\sigma^{54}$ binding and two UAS$_{LuxO}$ sites (*Figure 8—figure supplement 1*). The *sypG$_{AS}$* gene was cloned downstream of $P_{trc}$ and ectopically expressed in the Δ*luxO* Δ*sypG* mutant of *V. fischeri*. Using a GFP reporter for the promoter of *qrr1$_{AS}$* ($P_{qrr1AS}$), we found that induction of *sypG$_{AS}$* expression led to increased GFP fluorescence (*Figure 8*), which suggests that SypG$_{AS}$ can activate transcription of $P_{qrr1AS}$ and provides support that SypG-dependent expression of Qrr sRNAs can occur in other taxa within the *Vibrionaceae* family.

## Discussion

Quorum sensing enables cells within a bacterial population to collectively express traits (*Whiteley et al., 2017*; *Papenfort and Bassler, 2016*). The traits regulated by quorum sensing are often energetically costly, and bacteria have adapted to inhibit their expression under non-quorum conditions. In *Vibrionaceae*, inhibitory factors include Qrr sRNAs, which post-transcriptionally repress expression of a transcription factor that promotes the cellular responses to quorum sensing (*Papenfort and Bassler, 2016*). In this study, we discovered that *V. fischeri* has the potential to express Qrr1 even when responding to high concentrations of autoinducer (*Figure 9A*). Specifically, the bEBP SypG activates

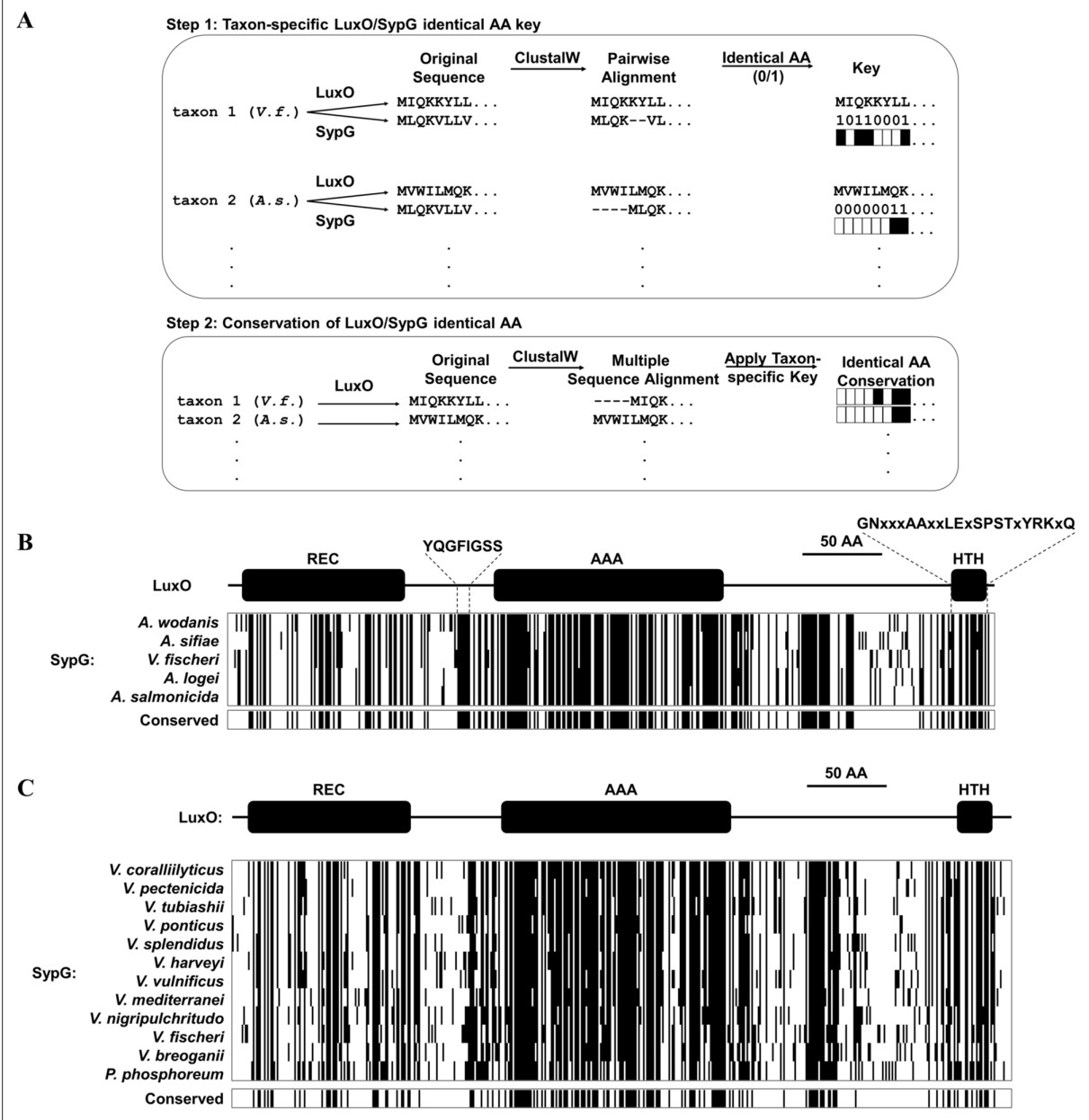

**Figure 7.** Conservation of SypG-LuxO structural features in the *Vibrionaceae* family. (**A**) Experimental design for visualizing the extent of identity conserved between the primary structures of LuxO and SypG homologs among different taxa. (**B**) Each block represents a multiple sequence alignment of LuxO homologs encoded within the indicated Fischeri clade members that has 481 amino acid positions including gaps. Positions marked by a black line indicate that the corresponding amino acid of the LuxO homolog is identical to that of SypG based on pairwise alignments. Shown below each block are the positions of amino acid identity that are conserved among the indicated taxa. (**C**) Each block represents a multiple sequence alignment of LuxO homologs encoded within the indicated *Vibrionaceae* members that has 489 amino acid positions including gaps. Positions marked by a black line indicate that the corresponding amino acid of the LuxO homolog is identical to that of SypG based on pairwise alignments. Shown below each block are the positions of amino acid identity that are conserved among the indicated taxa.

The online version of this article includes the following figure supplement(s) for figure 7:

**Figure supplement 1.** Conservation of Qrr1 within the Fischeri clade.

**Figure supplement 2.** Conservation of gene synteny for *luxO* and *sypG* within the Fisheri clade.

**Figure supplement 3.** Pairwise alignments of LuxO (top sequence) and SypG (bottom sequence) homologs encoded by indicated Fischeri clade members.

**Figure supplement 4.** Multiple sequence alignment of LuxO homologs encoded by Fischeri clade members.

*Figure 7 continued on next page*

*Figure 7 continued*

**Figure supplement 5.** Gene synteny associated with *luxO* in *Vibrionaceae* taxa with incomplete genomes.

**Figure supplement 6.** Conservation of *qrr* gene within *uvrB-luxO* intergenic region among *Vibrionaceae*.

**Figure supplement 7.** Gene synteny associated with *pepN* among select *Vibrionaceae*.

**Table 2.** LuxO and SypG homologs in *Vibrionaceae* clades.

| Clade | Taxon | LuxO homolog* | Accession | SypG homolog† | Accession | Identity‡ | Similarity‡ |
|---|---|---|---|---|---|---|---|
| Salinivibrio-Grimontia-Enterovibrio§ | *G. hollisae* | WP_005503370.1 | NZ_CP014056 | N.D. | — | N.A. | N.A. |
| Rosenbergii | *P. lutimaris* | WP_107348500.1 | NZ_SNZO01000002 | N.D. | — | N.A. | N.A |
| Profundum | *P. profundum* | WP_065814467.1 | NC_006370 | N.D. | — | N.A | N.A |
| Damselae | *Photobacterium damselae subsp. piscicida* | WP_086957069.1 | NZ_AP018045 | N.D. | — | N.A | N.A |
| Phosphoreum | *P. phosphoreum* | WP_045027808.1 | NZ_MSCQ01000001 | WP_105026695.1 | NZ_MSCQ01000001 | 237/499 (47.49%) | 310/499 (62.93%) |
| Fischeri | *V. fischeri* | WP_011261589.1 | NC_006840 | WP_011263835.1 | NC_006841 | 243/502 (48.41%) | 326/502 (64.94%) |
| Anguillarum | *V. anguillarum* | WP_026028983.1 | NC_022223 | N.D. | — | N.A | N.A |
| Rumoiensis | *V. rumoiensis* | N.D. | — | N.D. | — | N.A | N.A |
| Vulnificus | *V. vulnificus* | WP_011149911.1 | NC_014965 | WP_013571858.1 | NC_014965 | 247/508 (48.62%) | 320/508 (62.99%) |
| Diazotrophicus | *V. diazotrophicus* | WP_042486207.1 | NZ_POSL01000002 | N.D. | — | N.A | N.A |
| Gazogenes | *V. gazogenes* | WP_021019492.1 | NZ_CP018835 | N.D. | — | N.A | N.A |
| Porteresiae | *V. tritonius* | WP_068714228.1 | NZ_AP014635 | N.D. | — | N.A | N.A |
| Cholerae | *V. cholerae* | WP_001888250.1 | NC_002505 | N.D. | — | N.A | N.A |
| Halioticoli | *V. breoganii* | WP_065209630.1 | NZ_CP016177 | WP_065210697.1 | NZ_CP016178 | 228/513 (44.44%) | 298/513 (57.70%) |
| Splendidus | *V. splendidus* | WP_004734031.1 | NZ_CP031055 | WP_065205220.1 | NZ_CP031055 | 237/511 (46.38%) | 314/511 (61.45%) |
| Pectenicida | *V. pectenicida* | WP_125320437.1 | NZ_RSFA01000020 | WP_125322971.1 | NZ_RSFA01000107 | 237/501 (47.31%) | 321/501 (64.07%) |
| Scopthalmi | *V. ponticus* | WP_075650093.1 | NZ_AP019657 | WP_075649540.1 | NZ_AP019657 | 229/506 (45.26%) | 319/506 (63.04%) |
| Nereis | *V. nereis* | WP_061781622.1 | NZ_BCUD01000001 | N.D. | — | N.A | N.A |
| Orientalis | *V. tubiashii* | WP_038550519.1 | NZ_CP009354 | WP_004748949.1 | NZ_CP009354 | 242/497 (48.69%) | 319/497 (64.19%) |
| Coralliilyticus | *V. coralliilyticus* | WP_019275536.1 | NZ_CP048693 | WP_021455926.1 | NZ_CP048693 | 242/503 (48.11%) | 322/503 (64.02%) |
| Harveyi | *V. harveyi* | WP_005444697.1 | NZ_CP009467 | WP_050907635.1 | NZ_CP009467 | 244/522 (46.74%) | 320/522 (61.30%) |
| Nigripulchritudo | *V. nigripulchritudo* | WP_022603175.1 | NC_022528 | WP_022550524.1 | NC_022528 | 247/508 (48.62%) | 331/508 (65.16%) |
| Mediterranei | *V. mediterranei* | WP_062462808.1 | NZ_CP018308 | WP_088875891.1 | NZ_CP018308 | 236/503 (46.92%) | 318/503 (63.22%) |

*N.D. (not detected) indicates that the top hit from BLAST was a bEBP other than LuxO.

†N.D. (not detected) indicates that the top hit from BLAST was a bEBP other than SypG.

‡N.A. (not applicable) due to SypG homolog not detected.

§The Salinivibrio-Grimontia-Enterovibrio group is ancestrally related to the *Vibrionaceae* family and is included as an outgroup in this analysis.

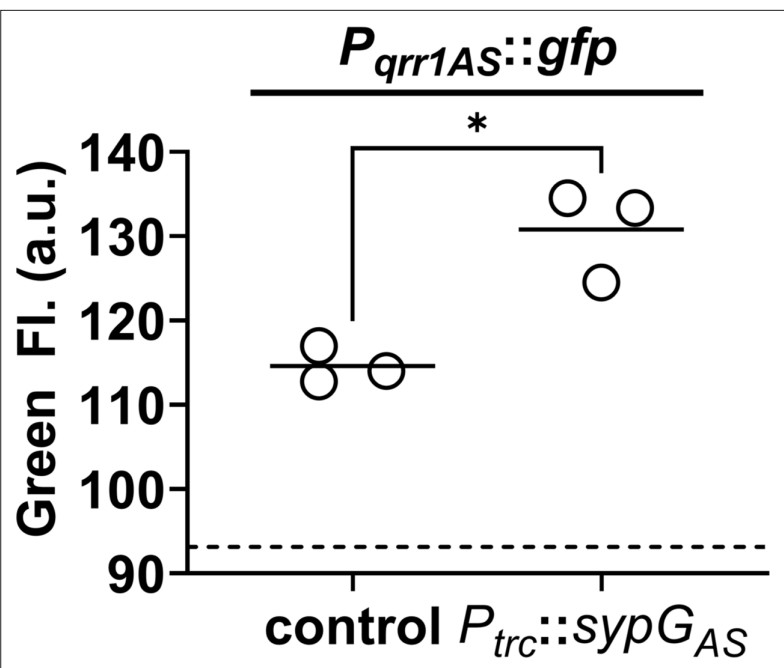

**Figure 8.** SypG-dependent transcription of *qrr1* for *A. salmonicida*. Green fluorescence of EDS008 (control) and EDS021 (Tn*7*::*sypG*<sub>AS</sub>) harboring pAGC003 (*P<sub>qrr1AS</sub>*::*gfp*). *N* = 3. Genotypes of both strains include Δ*luxO* and Δ*sypG* alleles, as well as *erm* integrated at the Tn*7* site. EDS008 harboring pVSV105 was used as a non-fluorescent control (dotted line). A paired *t*-test revealed significance between groups (*p = 0.0325). Experimental trials: 2.

The online version of this article includes the following source data and figure supplement(s) for figure 8:

**Source data 1.** Source data for *Figure 8*.

**Figure supplement 1.** Conservation of the *qrr1* locus among *V. fischeri* and *A. salmonicida*.

σ<sup>54</sup>-dependent transcription of *qrr1* in a manner that is independent of its primary bEBP LuxO. Transcriptional activation of *P<sub>qrr1</sub>* by SypG utilizes two UASs that overlap those sequences associated with LuxO-dependent activation. Together these findings reveal that *V. fischeri* has evolved to activate Qrr1 expression by either LuxO or SypG. The ability of SypG to activate σ<sup>54</sup>-dependent transcription of *P<sub>qrr1</sub>* in the presence of high autoinducer levels is significant because this regulatory link enables *V. fischeri* to bypass quorum sensing as a way to modulate the traits regulated by Qrr1.

When during the life cycle of *V. fischeri* would SypG-dependent activation of *P<sub>qrr1</sub>* be important? SypG activates σ<sup>54</sup>-dependent transcription of the *syp* locus, which enables *V. fischeri* to secrete polysaccharides that form an extracellular matrix (*Hussa et al., 2008*). Production of extracellular polysaccharides is necessary for *V. fischeri* to form the cellular aggregates on the surface of the light organ while initiating symbiosis (*Yip et al., 2005*; *Yip et al., 2006*; *Ray et al., 2013*). In culture, *syp*-dependent biofilm formation, which has been used to model the aggregation observed in vivo, depends on SypG activating expression of all four operons within the *syp* locus (*Ray et al., 2013*). While this study implicates *qrr1* as a member of the SypG regulon, its activation does not appear to contribute to biofilm formation (*Figure 5B*), which suggests that the Δ*qrr1* mutant can form aggregates prior to establishing symbiosis, in contrast to mutants containing deletions in other SypG-dependent genes. However, a Δ*qrr1* mutant shows fewer crypt populations relative to wild-type cells (*Figures 3 and 6*), which suggests that regulation by Qrr1 enhances the ability of *V. fischeri* to access a crypt space. Taken together, these findings support a model by which the environmental *V. fischeri* cells collected by the squid host express Qrr1 via SypG while forming *syp*-dependent aggregates along the light organ surface (*Figure 9B*). SypG-dependent activation of Qrr1 would prime cells to express certain traits that are enhanced by this sRNA, for example, cellular motility, precisely when the transition from the aggregate stage to light organ entry occurs. Notably, the ability of SypG to activate transcription of *P<sub>qrr1</sub>* makes the phosphorylation state of LuxO, and by extension, the corresponding quorum-sensing signaling pathway, irrelevant for expressing Qrr1 during this critical stage of initiating

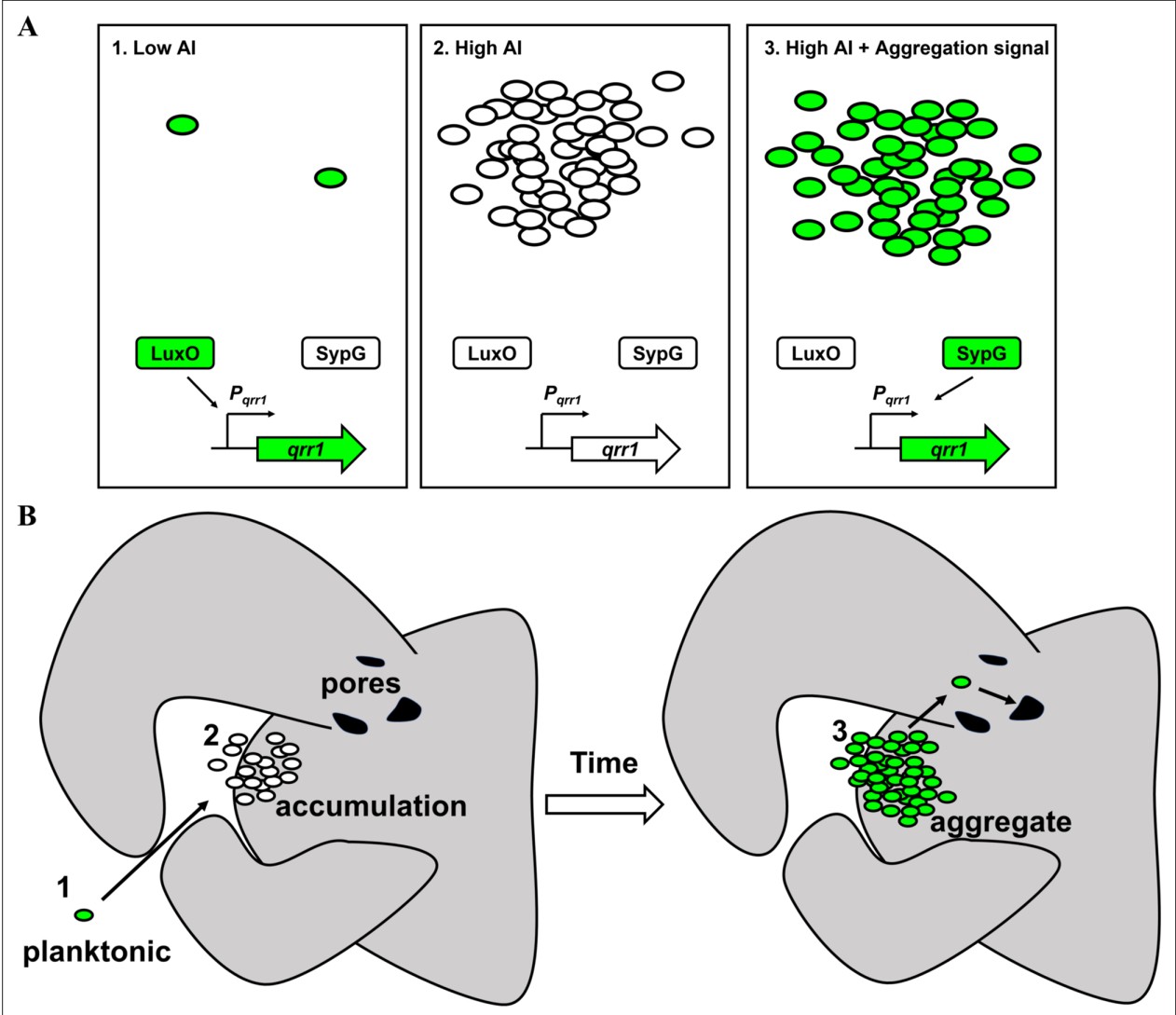

**Figure 9.** Model of dual bacterial enhancer binding protein (bEBP) control over Qrr1 expression in *V. fischeri*. (**A**) *Left,* cells in an environment with low autoinducer concentration, for example, low cell density, will express Qrr1 by activating LuxO through the quorum-sensing pathway. *Middle*, cells in an environment with high autoinducer concentration, for example, high cell density, will have low Qrr1 levels due to the inactive state of LuxO. *Right*, even under conditions of high autoinducer concentration, expression of Qrr1 can occur if SypG is activated by the aggregation pathway. (**B**) Model of initial entry of *V. fischeri* into the light organ. Planktonic cells within the environment express Qrr1 due to low autoinducer levels (panel A, box 1). Motion of the cilia associated with the appendages sweep bacteria into a stagnant zone, where they locally accumulate (panel A, box 2), which has the potential to lower Qrr1 expression. Within a few hours, the cells have formed aggregates that depend on SypG (panel A, box 3), which induces Qrr1 expression to prime cells for entry into the pores.

symbiosis. Interestingly, previous work has demonstrated that Qrr1 can be expressed under conditions of high cell density through the overexpression of SypK (*Miyashiro et al., 2014*), which is a putative oligosaccharide encoded by the *syp* locus. The current model is that SypK, which is predicted to localize to the inner membrane, activates $P_{qrr1}$ transcription by stimulating the LuxP/Q complex to trigger LuxO activity. More recently, it was also shown that a small molecule produced within RscS-induced wrinkled colonies promotes bioluminescence production (*Zink et al., 2021*), which suggests *V. fischeri* may feature additional connections between aggregation and quorum-sensing pathways. The finding that SypG can also activate $P_{qrr1}$ further expands the hypothesis that conditions that activate the *syp* locus, for example, when *V. fischeri* is initiating symbiosis, lead to the expression of Qrr1 as a mechanism to prime cells for host colonization.

Homologs of LuxO and Qrrs are encoded by most *Vibrionaceae* genomes, which underscores their biological significance in regulating the traits associated with quorum sensing. Over half of the

*Vibrionaceae* clades feature taxa that also encode SypG homologs with a high degree of amino acid identity/similarity to the corresponding LuxO homologs (*Table 2* and *Figure 7*). Such high similarity among primary structures is likely to promote higher-order structures within SypG that function similar to those of LuxO. For instance, the 1.6-Å resolution crystal structure derived from a partial-length construct of *V. angustum* LuxO (PDB entry 5EP0) features a linker region between the REC and AAA+ domains that sterically occludes nucleotide binding thereby preventing the ATP hydrolysis necessary for remodeling the RNAP–σ$^{54}$ complex to initiate transcription (*Boyaci et al., 2016*). A glycine conserved among all LuxO homologs both stabilizes this linker and occupies the active site, and, consistent with its predicted inhibitory role, substitution of the corresponding glycine in the LuxO homolog of *V. cholerae* with glutamate (G145E) results in increased LuxO activity (*Boyaci et al., 2016*). The analysis presented here shows that the primary structure of the linker is broadly conserved among the SypG homologs that are encoded by various *Vibrionaceae* members (*Figure 7*). To our knowledge, SypG represents the only other bEBP aside from LuxO predicted to contain this regulatory linker. Notably, examination of other *Vibrionaceae* clades did reveal some intriguing exceptions, for example, the position corresponding to a glycine within the linker is an asparagine in *V. splendidus* (N141) and an aspartate in *V. mediterranei* (D141). Both substitutions involve residues that are larger than glycine, which is the only residue that can fit within the active site of the *V. angustum* LuxO structure (*Boyaci et al., 2016*). Therefore, the SypG homologs of *V. splendidus* and *V. mediterranei* are likely to exhibit constitutive activity or feature other adaptations that accommodate for the altered linker structure. Future crystallographic and biochemical studies of these SypG homologs are necessary to test these possibilities. In addition, investigation into how each SypG homolog affects various traits in the corresponding taxon will provide insight into the various ecological roles of the *syp* locus among the *Vibrionaceae* family.

The REC domain within the N-terminal region of SypG implicates this bEBP as a response regulator that participates in two-component signaling. The signaling pathway that controls the phosphorylation status of SypG is extensive, with at least four hybrid histidine kinases (RscS, SypF, HahK, and BinK) that can affect SypG-dependent transcription of the *syp* locus (*Visick et al., 2021*). RscS is thought to phosphorylate the HPt domain within the C-terminus of SypF, which in turn phosphorylates D53 of SypG (*Norsworthy and Visick, 2015*). On the other hand, BinK is hypothesized to contribute to dephosphorylating SypG, due to the observation of the Δ*binK* mutant exhibiting higher SypG-dependent expression of the *syp* genes than WT cells (*Ludvik et al., 2021*). Previous work showed that increased biofilm production in strains lacking BinK depends on the HPt domain of SypF (*Thompson et al., 2018*), which highlights SypF as a potential phospho-donor for SypG in cells harboring a Δ*binK* allele. Our results that show elevated LuxO-dependent transcription of P$_{qrr1}$ in the Δ*binK* mutant (*Figure 4*) are consistent with higher levels of phosphorylated LuxO, which raises the possibility that BinK can dephosphorylate LuxO as well as SypG. How LuxO becomes phosphorylated in the Δ*binK* background remains untested, and future research is necessary to determine the full signaling pathway between BinK and regulation of *qrr1*. One prime candidate for the phospho-donor of LuxO is the HPt protein LuxU. Previous work has found that LuxU accelerates biofilm formation in cells overexpressing SypG (*Ray and Visick, 2012*), which provides a potential link to consider for additional studies of the signaling that occurs in a Δ*binK* background. Furthermore, in *V. cholerae*, LuxU can reverse phosphotransfer to multiple sensor kinases, which is thought to enable the VpsS to activate biofilm-related genes in a LuxO-dependent manner (*Shikuma et al., 2009*). Examining whether such promiscuity in phosphotransfer events occur among LuxU and the sensor kinases that regulate biofilm production in *V. fischeri* represents a worthwhile direction for future research.

Because bEBPs are critical for σ$^{54}$-dependent transcriptional activation (*Gao et al., 2020*) and their activity is usually controlled by signal transduction networks that sense environmental stimuli (*Bush and Dixon, 2012*), these specialized transcription factors also offer opportunities to engineer tightly controlled gene-regulatory modules for use in synthetic biology applications. For instance, the ability of LuxO and SypG to each activate transcription of P$_{qrr1}$ presented here resembles an OR logic gate that permits gene expression when either one or both of the bEBPs are active (e.g., *Figure 4D*). Molecular OR logic gates have been proposed as important components in engineering therapeutic bacteria that will deliver a drug when certain environmental conditions are satisfied (*Brophy and Voigt, 2014*). To our knowledge, evidence of different bEBPs activating the same gene has only been observed when the promoter exhibits distinct UASs that are specific for one or another bEBP. For

instance, the σ$^{54}$-dependent *dctA* gene of *Sinorhizobium meleloti* retains 20% transcriptional activity in a mutant lacking the *dctD* gene encoding the primary bEBP (**Wang et al., 1989**). This residual activity has been attributed to the bEBP NifA, for which potential UAS sites were identified within the *dctA* promoter region at sequences other than those associated with DctD binding (**Wang et al., 1989**; **Scholl and Nixon, 1996**). Despite SypG and LuxO utilizing the overlapping UASs upstream of P$_{qrr1}$, it appears that the σ$^{54}$-dependent promoters of the *syp* locus can be activated by SypG but not by LuxO (**Figure 4—figure supplement 3**). This discovery expands the utility of LuxO and SypG for synthetic biology with the *syp* promoters being appropriate for controlling gene expression with SypG alone. Consequently, determining the mechanism by which the *syp* promoters are insulated from LuxO in *V. fischeri* will not only reveal molecular insight into symbiont biology but will also further expand the utility of bEBPs in synthetic biology applications.

Full understanding of the structure–function relationship underlying the putative OR logic gate described in this study will require further investigation into the molecular details by which LuxO and SypG activate P$_{qrr1}$. For instance, determining how the HTH domain of each bEBP interacts with DNA will provide insight into whether competition between LuxO and SypG can affect dynamics of P$_{qrr1}$ activity under different environmental conditions. The genetic analysis presented here suggests that SypG recognizes each UAS associated with P$_{qrr1}$. However, we were unable to provide biochemical evidence that SypG binds to these sites, as our attempts to purify SypG for DNA-binding assays were stymied by protein insolubility and instability, which are problems that have been reported previously (**Hussa et al., 2008**; **Ray et al., 2013**). Activity of the bEBPs due to phosphorylation will likely also play a major role in how the regulatory module functions, for example, the SypG(D53E) variant can increase P$_{qrr1}$ activity even when LuxO is present (**Figure 4E**). Furthermore, the high degree of identity within the AAA+ domains may facilitate the assembly of LuxO–SypG heterohexamers with activity levels that are different from their homohexameric forms. Because the activity of each bEBP is linked to distinct signal transduction systems, this finding expands understanding of the environmental conditions that impact the cellular physiology of *V. fischeri*.

# Materials and methods

## Strains and plasmids

*V. fischeri* strains and plasmids used in this study are listed in **Table 3**. For cloning, *E. coli* strains Top10 and S17-1 λ pir were used. Primers used in the construction of strains and plasmids are listed in **Table 4**.

## Media and growth conditions

*V. fischeri* strains were grown at 28°C under aerobic conditions in LBS (Luria-Broth Salt) medium [1% (wt/vol) tryptone, 0.5% (wt/vol) yeast extract, 2% (wt/vol) NaCl, 50 mM Tris–HCl (pH 7.5)] or SWT (seawater-tryptone) medium (**Boettcher and Ruby, 1990**) with Instant Ocean (Blacksburg, VA) replacing seawater.

## Molecular biology

### Construction of mutants with deletion and/or *sypG*(D53E) alleles

Deletion alleles for *luxO*, *qrr1*, and *sypG* were introduced into strains by performing allelic exchange, as described previously (**Miyashiro et al., 2010**). Construction details of plasmids pTM235 and pTM238 that feature ΔluxO and Δqrr1, respectively, were described elsewhere (**Miyashiro et al., 2010**). Plasmid pEDR007, which encodes the deletion allele of *sypG* (ΔsypG) that lacks the codons encoding residues 49–478, was constructed by first amplifying by PCR from ES114 genomic DNA regions of homology upstream (primers sypF-KpnI-u1 and sypG-del-XbaI-l1) and downstream (primers sypG-del-XbaI-u1 and sypH-SacI-l1) of *sypG* and then cloning them into pEVS79 via KpnI/SacI. The deletion allele for *rpoN* (ΔrpoN) features the entire *rpoN* gene (1470 bp) replaced with the 78 bp FRT scar was introduced into strains by SOE PCR and recombineering mutagenesis (**Visick et al., 2018**) by generating regions of homology upstream (primers ES_rpoN Del Up F & R) and downstream (primers ES_rpoN Del Down F & R). To generate the strains with the *sypG*(D53E) allele, plasmid pDAT05 was used for allelic exchange and introduced into ES114 and MJM2251 as described previously (**Ludvik**

**Table 3.** Strains and plasmids used in this study.

| Strain name | Genotype | Reference |
| --- | --- | --- |
| ES114 | Wild-type *V. fischeri* | *Mandel et al., 2008* |
| DRO22 | ES114 Tn*5*::*binK* | This work |
| MJM2481 | ES114 Δ*binK* Tn*7*::*erm* | This work |
| TIM303 | ES114 Tn*7*::(P$_{qrr1}$::*gfp erm*) | *Miyashiro et al., 2010* |
| TIM313 | ES114 Tn*7*::*erm* | *Miyashiro et al., 2010* |
| TIM412 | ES114 Δ*binK* Tn*7*::(*binK erm*) | This work |
| MJM2251 | ES114 Δ*binK* | *Brooks and Mandel, 2016* |
| KRG004 | ES114 Δ*rpoN* | This work |
| KRG011 | ES114 Δ*binK* Δ*rpoN* | This work |
| EDR009 | ES114 Δ*binK* Δ*luxO* | This work |
| EDR013 | ES114 Δ*binK* Δ*luxO* Δ*sypG* | This work |
| EDR014 | ES114 Δ*binK* Δ*sypG* | This work |
| MJM4982 | ES114 *sypG*(D53E) | This work |
| MJM4983 | ES114 Δ*binK sypG*(D53E) | This work |
| EDS008 | ES114 Δ*luxO* Δ*sypG* Tn*7*::*erm* | This work |
| EDS010 | ES114 Δ*luxO* Δ*sypG* Tn*7*::(*lacI$^q$* P$_{trc}$::*sypG erm*) | This work |
| EDS015 | ES114 Δ*sypG* Tn*7*::(P$_{qrr1}$::*gfp erm*) | This work |
| SSC009 | ES114 Δ*sypK* Tn*7*::(P$_{qrr1}$::*gfp erm*) | This work |
| JHK007 | ES114 Δ*ainS* Δ*luxIR* P$_{lux}$-*luxCDABEG* | *Kimbrough and Stabb, 2013* |
| LFI1238 | Wild-type *Aliivibrio salmonicida* | *Hjerde et al., 2008* |
| EDS021 | ES114 Δ*luxO* Δ*sypG* Tn*7*::(*lacI$^q$* P$_{trc}$-*sypG$_{AS}$ erm*) | This work |
| MJM2255 | ES114 *rscS** Δ*binK* | *Brooks and Mandel, 2016* |
| MJM1198 | MJM1100 *rscS** | *Singh et al., 2015* |
| EDR010 | ES114 Δ*binK* Δ*qrr1* | This work |
| TIM305 | ES114 Δ*qrr1* | *Miyashiro et al., 2010* |
| SSC005 | ES114 Δ*qrr1* Tn*7*::*erm* | This work |
| TIM306 | ES114 Δ*luxO* | *Miyashiro et al., 2010* |
| TIM311 | ES114 Δ*luxO* Tn*7*::*erm* | *Miyashiro et al., 2010* |
| KRG016 | ES114 Δ*ainS* Δ*luxIR* Tn*7*::(P$_{trc}$::*gfp erm*) | This work |
| KRG018 | ES114 Δ*qrr1* Tn*7*::P$_{qrr1}$::*qrr1 erm* | This work |
| KRG019 | ES114 Δ*qrr1* Tn*7*::P$_{qrr1}$(G$_{-131}$T)::*qrr1 erm* | This work |
| KRG021 | ES114 Δ*qrr1* Tn*7*::*erm* | This work |
| **Plasmid name** | **Relevant genotype** | **Source** |
| pVSV105 | R6Kori *ori*(pES213) RP4 *oriT cat* | *Dunn et al., 2006* |
| pTM267 | pVSV105 *kan-gfp* P$_{tetA}$-*mCherry* | *Miyashiro et al., 2010* |
| pTM268 | pVSV105 P$_{qrr1}$-*gfp* P$_{tetA}$-*mCherry* | *Miyashiro et al., 2010* |
| pSCV38 | pVSV105 P$_{tetA}$-*yfp* P$_{tetA}$-*mCherry* | *Verma and Miyashiro, 2016* |
| pYS112 | pVSV105 P$_{proD}$-*cfp* P$_{tetA}$-*mCherry* | *Sun et al., 2016* |

*Table 3 continued on next page*

*Table 3 continued*

| Strain name | Genotype | Reference |
|---|---|---|
| pEDR003 | Region [(−373)–(+5)] of $P_{qrr1}$ cloned upstream of *gfp* reporter in pTM267 | This work |
| pEDR011 | Region [(−357)–(+5)] of $P_{qrr1}$ cloned upstream of *gfp* reporter | This work |
| pEDR012 | Region [(−262)–(+5)] of $P_{qrr1}$ cloned upstream of *gfp* reporter | This work |
| pEDR006 | Region [(−209)–(+5)] of $P_{qrr1}$ cloned upstream of *gfp* reporter | This work |
| pEDR009 | Region [(−106)–(+5)] of $P_{qrr1}$ cloned upstream of *gfp* reporter | This work |
| pEDR008 | Region [(−60)–(+5)] of $P_{qrr1}$ cloned upstream of *gfp* reporter | This work |
| pEDS007 | Region [(−373)–(+5)] of $P_{qrr1}$ with $G_{-97}T$ substitution cloned upstream of *gfp* reporter | This work |
| pEDS008 | Region [(−373)–(+5)] of $P_{qrr1}$ with $C_{-96}A$ substitution cloned upstream of *gfp* reporter | This work |
| pEDS009 | Region [(−373)–(+5)] of $P_{qrr1}$ with $A_{-95}C$ substitution cloned upstream of *gfp* reporter | This work |
| pEDS004 | Region [(−373)–(+5)] of $P_{qrr1}$ with $G_{-131}T$ substitution cloned upstream of *gfp* reporter | This work |
| pEDS005 | Region [(−373)–(+5)] of $P_{qrr1}$ with $C_{-130}A$ substitution cloned upstream of *gfp* reporter | This work |
| pEDS006 | Region [(−373)–(+5)] of $P_{qrr1}$ with $A_{-129}C$ substitution cloned upstream of *gfp* reporter | This work |
| pTM235 | pEVS79 Δ*luxO* | *Miyashiro et al., 2010* |
| pTM238 | pEVS79 Δ*qrr1* | *Miyashiro et al., 2010* |
| pDAT05 | pEVS79 *sypG*(D53E) | *Ludvik et al., 2021* |
| pEDR007 | pEVS79 Δ*sypG* | This work |
| pEVS79 | pBC SK (+) *oriT cat* | *Stabb and Ruby, 2002* |
| pEVS104 | | *Stabb and Ruby, 2002* |
| pEVS107 | R6Kori *oriT* mini-Tn7 *mob erm kan* | *McCann et al., 2003* |
| pTn7BinK | pEVS107 *binK* | *Brooks and Mandel, 2016* |
| pTM239 | pEVS107 $P_{qrr1}$-*gfp erm* | *Miyashiro et al., 2014* |
| pAGC003 | pEVS107 *lacI*$^q$ $P_{trc}$-*sypG*$_{As}$ | This work |
| pAGC004 | pTM267 $P_{qrr1AS}$-*gfp* | This work |
| pKG11 | pKV69 *rscS** | *Yip et al., 2006* |
| pKV69 | Mobilizable vector; *tet*$^R$ *cat* | *Visick and Skoufos, 2001* |
| pKRG040 | pEVS107 $P_{qrr1}$::*qrr1* | This work |
| pKRG041 | pEVS107 $P_{qrr1}(G_{-131}T)$::*qrr1* | This work |
| pVF_A1020P | pTM267 $P_{sypA}$::*gfp* | This work |
| pVF_A1035P | pTM267 $P_{sypP}$::*gfp* | This work |

**Table 4.** Primers used in this study.

| Primers | 5′ → 3′ |
| --- | --- |
| **Δ*sypG*** | |
| sypG-del-XbaI-l1 | CGGTCTAGATGTGGTGGATTCTTTTCCATAAATGCC |
| sypG-del-XbaI-u1 | GGCTCTAGAGTTAAGCCCGTCAACACTCT |
| sypF-KpnI-u1 | GGTACCGTTCTGGTTTAGGGTTAGCTATTTGTCA |
| sypH-SacI-l1 | GAGCTCCAGACAATAAAGAGGGGATGATAGC |
| **Δ*rpoN*** | |
| ES_rpoN Del Up F | CCTCAAGAAGCTTCTATTTTTAGAA |
| ES_rpoN Del Up R | TAGGCGGCCGCACTAAGTATGGTATTTAGCGATACCTTTTGTACATT |
| ES_rpoN Del Down F | GGATAGGCCTAGAAGGCCATGGTTAATGAAAAGGAAGTGTTATGCAA |
| ES_rpoN Del Down R | GATAGCTATCCCATTACCTATACCA |
| ***sypG*D53E sequencing** | |
| DAT_095 sypG fwd | CTACAGCAAGCCAGAAATGAAGCAG |
| DAT_096 sypG rev | GGGTGCCTTTTGATTGAATTAAGTTC |
| **pEDS003** | |
| sypG-pTrc-KpnI-u1 | GGTACCTTCGCTAGGTAAAACAGGATGTTA |
| sypG-pTrc-BsrGI-l1 | GGTGTACAGTAACCATATTTCATCATTCCGAT |
| **pAGC003** | |
| AS-KpnI-SypG-U1 | GGTACCTGCACAAGGCTTCACTA |
| AS-BsrGI-SypG-L1 | TGTACACAAAAGCCATACCTCAAAAG |
| **pEDR003** | |
| qrr1-prom-XmaI-u2 | GGCCCGGGCAGCCAACACATCAAAACCTGTCA |
| qrr1-prom-XbaI-l2 | GGTCTAGAACTAGTGGTCAATATACCTATTGCAGGGAG |
| **pEDR006** | |
| qrr1-prom-XmaI-u3 | GGCCCGGGGGTATCATCAAATCCAACTTGAGGG |
| qrr1-prom-XbaI-l2 | GGTCTAGAACTAGTGGTCAATATACCTATTGCAGGGAG |
| **pEDR008** | |
| qrr1-XmaI-reg1-u1 | GCGCCCGGGGGCTTATTTAGCTTATTTTTACG |
| gfp-XhoI-l1 | TACTCGAGTTTGTGTCCGAGAATGTTTCCATC |
| **pEDR009** | |
| qrr1-XmaI-reg2-u1 | CCGCCCGGGACGCAATTTGCAAAATGC |
| gfp-XhoI-l1 | TACTCGAGTTTGTGTCCGAGAATGTTTCCATC |
| **pEDR010** | |
| qrr1-XmaI-reg5-u1 | GGCCCGGGCAATATCAAAACCTAACGGG |
| gfp-XhoI-l1 | TACTCGAGTTTGTGTCCGAGAATGTTTCCATC |
| **pEDR011** | |
| qrr1-XmaI-reg7-u1 | GGCCCGGGACCTGTCATGTCAGGC |
| gfp-XhoI-l1 | TACTCGAGTTTGTGTCCGAGAATGTTTCCATC |
| **pEDR012** | |
| qrr1-XmaI-reg4-u1 | CCGCCCGGGGCAGTATCTTCTACCATTAATAAA |

*Table 4 continued on next page*

*Table 4 continued*

| Primers | 5′ → 3′ |
|---|---|
| gfp-XhoI-l1 | TACTCGAGTTTGTGTCCGAGAATGTTTCCATC |
| **pEDS004, pKRG041** | |
| qrr1-prom-SDM-G243T-u1 | TAAAAATGCGGTTGATATTTTCATTATGCAATCAGGATTCG |
| qrr1-prom-SDM-G243T-l1 | CGAATCCTGATTGCATAATGAAAATATCAACCGCATTTTTA |
| **pEDS005** | |
| qrr1-prom-SDM-C244A-u1 | AAAAATGCGGTTGATATTTGAATTATGCAATCAGGATTCGC |
| qrr1-prom-SDM-C244A-l1 | GCGAATCCTGATTGCATAATTCAAATATCAACCGCATTTTT |
| **pEDS006** | |
| qrr1-prom-SDM-A245C-u1 | AAAATGCGGTTGATATTTGCCTTATGCAATCAGGATTCGCA |
| qrr1-prom-SDM-A245C-l1 | TGCGAATCCTGATTGCATAAGGCAAATATCAACCGCATTTT |
| **pEDS007** | |
| qrr1prom-mut_G277T_u1 | GGATTCGCAAAACGCAATTTTCAAAATGCAAAAAAGGATG |
| qrr1prom-mut_G277T_l1 | CATCCTTTTTTGCATTTTGAAAATTGCGTTTTGCGAATCC |
| **pEDS008** | |
| qrr1prom-mut_G278A_u1 | GATTCGCAAAACGCAATTTGAAAAATGCAAAAAAGGATGAC |
| qrr1prom-mut_G278A_l1 | GTCATCCTTTTTTGCATTTTTCAAATTGCGTTTTGCGAATC |
| **pEDS009** | |
| qrr1prom-mut_G279C_u1 | CGCAAAACGCAATTTGCCAAATGCAAAAAAGGATG |
| qrr1prom-mut_G279C_l1 | CATCCTTTTTTGCATTTGGCAAATTGCGTTTTGCG |
| **pKRG040** | |
| Qrr1-SpeI-u1 | CCGGACTAGTTAGTTAGTTATTGATTTTAA |
| Qrr1-KpnI-l1 | CCGGGGTACCCAGCCAACACATCAAAACCT |
| **pAGC003** | |
| AS-KpnI-SypG-U1 | GGTACCTGCACAAGGCTTCACTA |
| AS-BsrGI-SypG-L1 | TGTACACAAAAGCCATACCTCAAAAG |
| **pAGC004** | |
| AS-Qrr1-XmaI-U1 | CCCGGGGTCCAGTCATATCCGGCAAGC |
| AS-Qrr1-XbaI-L1 | TCTAGAGGTCACTATACATATAGCAGAG |

*et al., 2021*), with the resulting strains validated by sequencing the *sypG* locus amplified by primers DAT_095 sypG fwd and DAT_096 sypG rev.

## Chromosomal integration

Plasmids pEVS107, pTn7binK, pTM239, pEDS003, pKRG040, pKRG041, and pAGC003 were used to introduce genetic content in single copy into the chromosome at the Tn7 site, as described elsewhere (*McCann et al., 2003*). Plasmid pEDS003 was constructed by first amplifying *sypG* by PCR from ES114 genomic DNA (primers sypG-pTrc-KpnI-u1 and -BsrGI-l1) and cloning the product downstream of the P$_{trc}$ promoter in pTM318 via KpnI/BsrGI. Plasmid pKRG040 was generated by amplifying *qrr1* and its native promoter region (P$_{qrr1}$) from ES114 genomic DNA using PCR (primers Qrr1-SpeI-u1 and Qrr1-KpnI-l1) and cloning into the pEVS107 vector via SpeI/KpnI. The amplicon within the pKRG040 plasmid was subjected to site-directed mutagenesis (described below) to generate plasmid pKRG041.

Plasmid pAGC003 was constructed in similar fashion using the amplicon (primers AS-KpnI-SypG-U1 and -BsrGI-SypG-L1) generated from LFI1238 genomic DNA.

## Promoter transcriptional reporters

Plasmids pEDR003 and pEDR006 were constructed by amplifying the $P_{qrr1}$ region from ES114 genomic DNA by PCR (primers qrr1-prom-XmaI-u2 and -XbaI-l2 and -XmaI-u3 and XbaI-l2, respectively) and cloning the products upstream of *gfp* in pTM267 via XmaI/XbaI. Reporter plasmids pEDR011, pEDR010, pEDR012, pEDR009, and pEDR008, which contain truncated $P_{qrr1}$ regions, were constructed by amplifying from pEDR003 by PCR (reverse primer gfp-XhoI-l1 and respective forward primers qrr1-XmaI-reg7-u1, -reg5-u1, -reg4-u1, -reg2-u1, and -reg1-u1) and cloning the resulting products into pTM267 via XmaI/XhoI. Plasmid pAGC004, which contains the $P_{qrr1AS}$-*gfp* reporter, was constructed by amplifying the $P_{qrr1AS}$ region from LFI1238 genomic DNA by PCR (primers AS-Qrr1-XmaI-U1 and -XbaI-L1) and cloning the product into pTM267 via XmaI/XbaI.

### Site-directed mutagenesis

The amplicon generated for pEDR003 (primers qrr1-prom-XmaI-u2 and XbaI-l2), which contains the $P_{qrr1}$ region, was cloned into pCR-blunt (Thermo Fisher) and used as a template for site-directed mutagenesis. Primers listed for pEDS004, pEDS005, pEDS006, pEDS007, pEDS008, and pEDS009 were used to conduct PCR with Pfu Ultra (Agilent) for 18 cycles. The reaction was subjected to DpnI digest, transformed by electroporation into Top10 *E. coli* cells, and validated by sequencing before subcloning into pTM267 via XmaI/XbaI. The plasmid pKRG041 was generated with a similar technique using the amplicon within pKRG040, which contains $P_{qrr1}$-*qrr1*, as a template. Primers for the mutagenesis are listed in *Table 4*. After validating the mutagenesis via sequencing, the insert was subcloned into pEVS107 via SpeI/KpnI, and transformed to chemically competent EC100pir+ cells.

## Promoter-activity spotting assays

Starter cultures of *V. fischeri* strains were grown overnight in LBS broth supplemented with 2.5 μg/ml chloramphenicol. For each culture, a 1-ml sample was prepared by adjusting its turbidity to an $OD_{600}$ equivalent to 1.0. To initiate the assay, a 2.5-μl sample of the cell suspension was placed onto the surface of LBS agar supplemented with 2.5 μg/ml chloramphenicol (and 150 μM isopropyl ß-D-1-thiogalactopyranoside [IPTG] where indicated) and incubated at 28°C. After 24 hr, the spots were examined at ×4 magnification using an SZX16 fluorescence dissecting microscope (Olympus) equipped with an SDF PLFL ×0.3 objective and both GFP and mCherry filter sets. Images of green fluorescence and red fluorescence of the spot were captured using an EOS Rebel T5 camera (Canon) with the RAW image format setting. Image analysis was performed using ImageJ, v. 1.52a (NIH) as follows. First, images were converted to RGB TIFF format using the DCRaw macro, with the following settings selected: use_temporary_directory, white_balance = [Camera white balance], do_not_automatically_brighten, output_colorspace = [sRGB], read_as = [8-bit], interpolation = [High-speed, low-quality bilinear], and half_size. For each spot, the green channel of the green fluorescence image was used for quantifying GFP fluorescence, and the red channel of the mCherry fluorescence image was used for quantifying mCherry fluorescence. The region of interest (ROI) corresponding to the spot was identified in the red channel by thresholding, and this ROI was used to determine the mean red and green fluorescence levels for each spot. A non-fluorescent sample (pVSV105/ES114) was used to determine the levels of cellular auto-fluorescence. A one-way analysis of variance with Dunnett's multiple comparisons test was performed to determine whether groups were significantly different than the non-fluorescent control group. The fold change in fluorescence between two groups was determined by first subtracting auto-fluorescence levels from each group mean and then calculating the ratio of the differences.

## Bioluminescence assay

Starter LBS cultures of the indicated *V. fischeri* strains were grown overnight and then subcultured 1:100 into SWT medium. At indicated time points, turbidity ($OD_{600}$) and luminescence (RLUs) measurements were collected using a Biowave CO8000 Cell Density Meter and a Promega GloMax 20/20 luminometer, respectively. Specific luminescence for each sample was calculated by normalizing each luminescence measurement with the corresponding turbidity measurement.

## Light organ colonization assay

Starter cultures of the indicated *V. fischeri* strains were initiated with LBS medium supplemented with 2.5 µg/ml chloramphenicol for plasmid maintenance. Following overnight incubation, culture samples were normalized to an $OD_{600}$ = 1.0 and diluted 1:100 in fresh medium. After cultures had reached $OD_{600}$ = 1.0, they were diluted into filter-sterilized Instant Ocean seawater (FSSW). For each group, freshly hatched juvenile squid (*E. scolopes*) derived from wild-caught adult animals collected in Oahu, HI and maintained in a mariculture facility (*Cecere and Miyashiro, 2022*) were exposed collectively to an inoculum mixed evenly with cell suspensions of the indicated *V. fischeri* strains. The total cellular abundance and ratio of strain types in each inoculum were determined by plating serial dilutions and using a fluorescence dissecting microscope to count the resulting colonies exhibiting YFP and CFP fluorescence. Inoculum levels ranged between $4 \times 10^3$ and $1 \times 10^5$ CFU/ml and corresponding ratios were not significantly different from 1.0. After being exposed to the inoculum for 3.5 hr, squid were washed three times in FSSW and then housed individually in vials containing 4 ml FSSW. Each day, squid were transferred to vials containing fresh FSSW. At 44 hr post-inoculation, squid of each group were combined and anesthetized on ice with 5% ethanol/FSSW and then fixed as a group in marine phosphate buffer containing 4% paraformaldehyde at 4°C. After 24 hr, squid were washed four times with marine phosphate buffer and dissected to reveal the light organ. For each light organ, images of YFP, CFP, and DIC were acquired using a 780 NLO confocal microscope (Carl Zeiss AG, Jena, Germany) equipped with a ×10 water lens and pinholes set to maximum to mimic epi-fluorescence conditions. The YFP and CFP fluorescence images of each light organ were visually examined in conjunction with the DIC image to score each region associated with a crypt space for fluorescence signal. Animal experiments were performed using protocol approved by the Institutional Animal Care and Use Committee at Penn State University (#PROTO202101789).

## Aggregation assay

Starter LBS + 2.5 µg/ml chloramphenicol cultures of indicated strains harboring pSCV38 were diluted 1:100 into fresh medium and grown to an $OD_{600}$=1.0. Cells were washed twice with each step consisting of centrifugation at $5000 \times g$ for 2 min, aspiration of the supernatant, and resuspension of the pellet into FSSW. The assay was initiated by exposing squid as a group to $5.0 \times 10^6$ CFU/ml. After 3.5 hr, squid were anesthetized by placing on ice for 5 min and then exposing them to 3% ethanol/FSSW for at least 15 min. The light organ was exposed by dissection with forceps and imaged using fluorescence microscopy. Each light organ was scored for aggregates by assessing the green fluorescence image of each side for the presence of a particle. Aggregate size was determined using the default IsoData auto-threshold method of the threshold tool in ImageJ to generate a binary image from the green fluorescence image, which was then subjected to the analyze particles command, with pixel^2 size range set to 10-infinity, to measure the area of each particle.

## Wrinkled-colony assay

Starter cultures of *V. fischeri* strains harboring either pKG11 (*rscS\**) or pKV69 (vector) were grown overnight in LBS broth supplemented with 2.5 µg/ml chloramphenicol. For each culture, a 1-ml sample was prepared by adjusting its turbidity to an $OD_{600}$ equivalent to 1.0. To initiate the assay, a 2.5-µl sample of the cell suspension was placed onto the surface of LBS agar supplemented with 2.5 µg/ml chloramphenicol and incubated at 25°C. After 24 hr, the spots were examined at ×4 magnification using an SZX16 fluorescence dissecting microscope (Olympus) equipped with an SDF PLFL ×0.3 objective and either a GFP filter (green fluorescence) or no filter (brightfield). Images were acquired as described in the promoter-activity spotting assay above.

## Statistical analysis

Except where indicated in the figure legend, experiments were performed at least three times. We define biological replicates as biologically distinct samples showing biological variation, and technical replicates as repeated measurements of a single sample. The number of biological replicates (*N*) is listed in figure legends. All statistical tests were performed in GraphPad Prism version 9.3.1 and listed in figure legends. Justification for statistical tests was determined by performing a Shapiro–Wilk test for normality on group data (or log-transformed data). Experiments in which normality failed (p-value ≥0.05) were statistically analysed using nonparametric statistical tests.

## Gene synteny analysis

Analysis of gene synteny was performed by downloading GenBank files containing the indicated sequences from NCBI and subjecting them to the progressiveMauve algorithm (*Darling et al., 2010*), which identifies locally colinear blocks (LCBs) that are genomic segments that are conserved independent of rearrangements due to recombination. The following parameters were selected for each run: default seed weight, determine LCBs, full alignment, iterative refinement, and sum-of-pairs LCB scoring.

## Protein alignments

Protein sequences were downloaded as FASTA format from NCBI and pasted directly into the Alignment Explorer tool of MEGA X (*Kumar et al., 2018*). Alignments were performed using ClustalW, with Gap Opening Penalty = 10.00 and Gap Extension Penalty = 0.10 and 0.20 for pairwise and multiple sequence alignments, respectively. Alignments were exported in.fas format. For pairwise alignments, the identity and similarity values were determined using the Ident and Sim program of the Sequence Manipulation Suite (SMS) (*Kumar et al., 2018*). Alignment displays were generated using the Color Align Conservation program of SMS, with similar amino acid groups defined as GAVLI, FYW, CM, ST, KRH, DENQ, P.

To visualize the positions of residues that are identical between LuxO and SypG homologs across a set of taxa, a multisequence alignment of the LuxO homologs encoded by those taxa was first generated. Each pairwise alignment was used to generate a key that indicates for each residue in LuxO whether the corresponding position within the alignment contains an amino acid that is identical (labeled as 1) or not identical (labeled as 0). The keys from the pairwise alignments were used to replace the amino acid letters within the LuxO multisequence alignment with the identical/not identical values. Using Excel, cells containing a 1 were formatted with black fill and those cells containing a 0 were formatted with white fill. The resulting table grid was used to generate the corresponding image shown in this report. The consensus array was generated in similar fashion after determining which positions across rows within the alignment contained a value of 1.

## Material availability statement

Reasonable requests for plasmids and strains can be made to corresponding author (TIM).

## Acknowledgements

This work was supported by National Institute of General Medical Sciences Grants R01 GM129133 (to TIM) and R35 GM148385 (to MJM), Howard Hughes Medical Institute Gilliam Fellowship (to EDS and TIM), and National Institute of Allergy and Infectious Diseases Fellowship F32 AI 147543 (to KRG).

## Additional information

### Funding

| Funder | Grant reference number | Author |
| --- | --- | --- |
| National Institute of General Medical Sciences | R01 GM129133 | Tim I Miyashiro |
| National Institute of General Medical Sciences | R35 GM148385 | Mark J Mandel |
| Howard Hughes Medical Institute | | Ericka D Surrett |
| National Institute of Allergy and Infectious Diseases | F32 AI147543 | Kirsten R Guckes |

The funders had no role in study design, data collection, and interpretation, or the decision to submit the work for publication.

## Author contributions

Ericka D Surrett, Conceptualization, Data curation, Formal analysis, Funding acquisition, Validation, Investigation, Visualization, Methodology, Writing – original draft, Writing – review and editing; Kirsten R Guckes, Shyan Cousins, Formal analysis, Investigation, Methodology; Terry B Ruskoski, Andrew G Cecere, Investigation; Denise A Ludvik, Investigation, Methodology; C Denise Okafor, Resources, Investigation, Visualization; Mark J Mandel, Resources, Supervision, Investigation, Methodology, Writing – review and editing; Tim I Miyashiro, Conceptualization, Resources, Data curation, Software, Formal analysis, Supervision, Funding acquisition, Validation, Investigation, Visualization, Methodology, Writing – original draft, Project administration, Writing – review and editing

## Author ORCIDs

Kirsten R Guckes http://orcid.org/0000-0002-0929-3351
Denise A Ludvik http://orcid.org/0000-0001-7280-7362
C Denise Okafor http://orcid.org/0000-0001-7374-1561
Mark J Mandel http://orcid.org/0000-0001-6506-6711
Tim I Miyashiro http://orcid.org/0000-0002-5016-1641

## Ethics

Animal experiments were performed using protocol approved by the Institutional Animal Care and Use Committee at Penn State University (#PROTO202101789).

## Decision letter and Author response

Decision letter https://doi.org/10.7554/eLife.78544.sa1
Author response https://doi.org/10.7554/eLife.78544.sa2

---

# Additional files

## Supplementary files

• MDAR checklist

## Data availability

Numerical data used to generate graphs in Figures 2, 3, 4, 5, 6, and 8 are present within corresponding Source Data files. Alignments used in generating Figure 7 are located in the Supporting file.

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
