## [Editor Report]

The authors present a rigorous and valuable study in which they identify the role of the conserved bacterial enhancer binding protein (bEBP) SypG in regulation of the Qrr1 small RNA, a key regulator of Vibrio fischeri bioluminescence production and squid colonization. The research design and methods were convincing and thorough, leading to compelling conclusions that are broadly relevant to both the quorum sensing and Vibrio-squid symbiosis fields.

---

## [Decision Letter]

**Decision letter after peer review:**

Thank you for submitting your article "Two enhancer binding proteins activate σ^54^ -dependent transcription of a quorum regulatory RNA in a bacterial symbiont" for consideration by *eLife*. Your article has been reviewed by 3 peer reviewers, one of whom is a member of our Board of Reviewing Editors, and the evaluation has been overseen Wendy Garrett as the Senior Editor. The reviewers have opted to remain anonymous.

Essential revisions:

The reviewers agree that the manuscript has an important potential impact in the field but that the following points should be addressed prior to resubmission to *eLife*:

1. What is the connection between BinK and SypG? Why RcsS* is needed in some experiments but not all?

2. Is SypG phosphorylation/dephosphorylation important and is it controlled by BinK?

3. Is SypG a direct regulator of Qrr1?

4. If SypG is a direct regulator, does SypG compete with LuxO on Pqrr1 binding?

---

## [Author Response]

Essential revisions:The reviewers agree that the manuscript has an important potential impact in the field but that the following points should be addressed prior to resubmission to eLife:1. What is the connection between BinK and SypG? Why RcsS* is needed in some experiments but not all?

We have added a figure panel (Figure 2C) and details in the text to highlight the signaling pathway associated with BinK, SypG, and RscS. Briefly, RscS is a key phosphodonor at the top of the pathway, so in the previous manuscript version, approaches to stimulate the SypG pathway independent of BinK (or Δ*binK* allele) were accomplished with the RscS* allele. In the revised manuscript, the SypG(D53E) variant provides independent validation of this approach.

2. Is SypG phosphorylation/dephosphorylation important and is it controlled by BinK?

In the revised manuscript, Figure 2C should clarify the importance of phosphorylation/dephosphorylation, as well as the role of BinK in this process. Efforts are underway in the Mandel Laboratory to establish the mechanism by which BinK acts to regulate SypG, but those approaches are challenging and beyond the scope of this manuscript. However, by using the phosphomimetic SypG variant in the new experiments here (Figure 4E), we were able to interrogate the role of SypG activation in promoting P*_qrr1_* expression.

3. Is SypG a direct regulator of Qrr1?

Unfortunately, despite multiple attempts with different tags (e.g., N-/C-terminal His/TAP tags), we have not had success in generating a construct that would permit these studies. Difficulty with SypG has been reported in previous studies, and we have added sections in the text to highlight this limitation of our report. Nonetheless, we have added substantial data described above using the phosphomimetic SypG variant to probe the potential role of phosphorylation.

4. If SypG is a direct regulator, does SypG compete with LuxO on Pqrr1 binding?

Due to the complications associated with item 3, we were unable to address this question. However, the data associated with point mutations within the P*_qrr1_* region have identified nucleotides that are important for both regulators or only SypG. These findings led to the construction of a new strain that enabled us to isolate the SypG-Qrr1 pathway and demonstrate its importance during colonization. Furthermore, that two of the attempted substitutions in each UAS attenuated activation of P*_qrr1_* by both regulators strongly suggests that they compete for P*_qrr1_*.